# Adsorption of Wormlike Chains onto Partially Permeable Membranes

**DOI:** 10.3390/polym15010035

**Published:** 2022-12-22

**Authors:** Alexander Semenov, Irina Nyrkova

**Affiliations:** Institut Charles Sadron, CNRS-UPR 22, University of Strasbourg, CEDEX 2, 67034 Strasbourg, France

**Keywords:** adsorption, polymers, membranes

## Abstract

Reversible adsorption of a single stiff wormlike macromolecule to flat membranes with various permeabilities is considered theoretically. It is shown that the adsorbed layer microstructure is significantly different from either a flexible chain or a stiff chain adsorption at a solid surface. Close to the critical point, the adsorbing wormlike chain forms a strongly anisotropic proximal layer near the membrane in addition to a nearly isotropic distal layer. The proximal layer is characterized by the algebraic monomer concentration profile, $c\left(x\right)\propto {\left|x\right|}^{-\beta }$, due to the self-similar distribution of aligned polymer loops. For a perfectly penetrable membrane, β=1 which is different from β=4/3 obtained for semiflexible chain adsorption at a solid surface. Moreover, we establish that the critical exponent for a partially permeable membrane depends on its properties (porosity *w*) and propose an asymptotically exact theory (based on the generalized Edwards equation) predicting this dependence, β=β(w). We also develop a scaling theory elucidating, in particular, an intricate competition of loops and tails in both proximal and distal sublayers.

## 1. Introduction

Polymer adsorption is a remarkable phenomenon relevant to many applications [1,2,3,4]. It is routinely used to modify surface properties by coating with adsorbed polymer layers for protection, lubrication and adhesion, and to tune the physical properties of interfacial functional layers. Adsorption from a solution can efficiently impart the stabilization of colloidal and nano-particles (including protein complexes), preventing their aggregation [1,2,5,6,7]. Single-chain adsorption is involved in many biological processes [8,9,10,11].

Both equilibrium properties and kinetics of flexible polymer adsorption have been studied theoretically since long ago [3,4]. Typically, polymer adsorption is caused by short-range monomer–surface interactions (of strength *u*). For flexible polymers, it was established that single-chain adsorption is akin to a critical phenomenon (a second-order phase transition), where the order parameter *p* (the fraction of monomers in contact with the attractive surface) changes in a continuous way, becoming nonzero beyond the critical threshold $u^{*}$ (at $u>u^{*}$) in the limit N→∞, where *N* is the number of monomers in the chain [4,12]. Exactly at $u=u^{*}$, the number of surface contacts $n_{c}=Np$ scales as
(1)$$n_{c}\propto N^{\phi }$$
where ϕ is the crossover exponent [4,12,13]. The polymer concentration profile c(x) at $u=u^{*}$ near the surface follows the critical scaling law, $c\left(x\right)\propto x^{-\beta }$, where *x* is the distance to the surface. For ϕ<1, the exponent β is also smaller than 1 and is related to ϕ: $\beta =1-\frac{1-\phi }{\nu }$, where ν is the exponent defining the size (gyration radius $R_{G}$) of an isolated coil, $R_{G}\propto N^{\nu }$. In a good solvent, ϕ≈0.485 [14,15,16] and ν≈0.588 [17], so β≈0.124, while in Θ-solvent (or marginal solvent) polymer coils follow the ideal-chain statistics (apart from logarithmic corrections) [4], so ν=0.5, ϕ=0.5 and β=0.

One of the most important characteristics of macromolecules is their backbone stiffness, which is related to the correlation length $l_{p}$ of the chain orientation along its contour. Stiff polymers have large persistence length $l_{p}$, which is much longer than their thickness *d*: $l_{p}/d\gg 1$. Many physical and, in particular, adsorption properties of such semirigid (or semiflexible) macromolecules are significantly different [18,19,20,21,22,23] from those of flexible chains (whose $l_{p}$ is comparable to *d*). The family of stiff polymers is wide: it includes synthetic helical polymers (such as poly-γ-benzyl-L-glutamate), polyelectrolytes, and important biological polymers, such as double-strand DNA, polypeptides and proteins, microtubules, F-actin and other protein polymers (biofilaments). By far, the most used statistical model of semirigid polymers is the wormlike chain (Kratky–Porod) model [8,18,21,24]. The model is also applicable to biopolymers [25]. For example, as pointed out in ref. [10], on length scales smaller than the Kuhn segment length, accessible by scanning force microscopy, conformations of a single DNA molecule were shown [26] to be appropriately described by the wormlike chain model. (Some recently reported deviations [23] from the wormlike chain behavior for some biopolymers only confirm the central role of the Kratky–Porod model). This model is adopted here; it implies that the orientational correlations along the chain contour decay exponentially with the curvilinear (contour) distance *s*:(2)$$\langle \underline{n}\left(s+s^{\prime }\right)\cdot \underline{n}\left(s^{\prime }\right)\rangle =e^{-s/l_{p}}$$
where $\underline{n}\left(s^{\prime }\right)$ and $\underline{n}\left(s+s^{\prime }\right)$ are unit vectors tangential to the backbone at points $s^{\prime }$ and $s+s^{\prime }$, respectively. The mean-square end-to-end distance of a *long* persistent (wormlike) chain is
(3)$$R^{2}≃lL$$
where $l=2l_{p}$ is Kuhn segment of the chain, and L≫l is its contour length. In the opposite regime, L≪l, we trivially have R≃L.

Back in 2002, we considered the adsorption of a semiflexible wormlike chain at a solid surface [21]. It was shown that while the adsorption transition is continuous, it becomes progressively sharper as the chain rigidity increases. In particular, the fraction of adsorbed monomers (in a thin layer of thickness ∼Δ within the surface attractive potential range) jumps from nearly 0 to almost 1 in a narrow temperature range (around the critical point) whose relative width is proportional to ${\left(\Delta /l\right)}^{4/3}$, where *l* is the Kuhn segment of the chain. In this respect, the adsorption transition becomes similar to a first-order phase transition. It was also shown [21] that at the critical point, the polymer concentration follows the specific scaling law, $c\left(x\right)\propto x^{-4/3}$ for Δ≲x≲l.

In the present paper, we analyze the adsorption of wormlike macromolecules to a thin free-standing film, such as a membrane or a giant surfactant vesicle, or a lipid bilayer. It may seem that adsorption onto a film is similar to the solid surface case. However, the main difference from the previously considered case is that now, we take into account that the film can be (at least, partially) penetrable for the adsorbing chain.

In the next section, we consider the adsorption of a short polymer chain (with L≲l) to a perfectly penetrable membrane. First, we present a scaling argument predicting the critical strength $u^{*}$ of polymer/membrane attraction required for adsorption. Next we develop a scaling theory for polymer statistics at the adsorption threshold $(u=u^{*}$) and slightly below $(u<u^{*}$) or above it. The *u*-dependencies of the energy *E* (which is proportional to the fraction *p* of adsorbed polymer segments located within the membrane attraction range) and the adsorption free energy *F* are considered there as well. In Section 3, we develop a quantitative approach to analyze the adsorption microstructure, using the generalized Edwards equation. The theory is focused on the proximal layer (of thickness ≲l) formed by long chains at $u=u^{*}$. A more general case, $u\neq u^{*}$, is considered in Section 4 based on scaling arguments. The relevant experimental systems are discussed in Section 5. The adsorption of a semiflexible wormlike polymer to partially penetrable porous membranes (solid films with holes) is then studied in detail in Section 6. The mains results are discussed and summarized in Section 7 and Section 8, respectively.

## 2. Perfectly Penetrable Free-Standing Film

### 2.1. Adsorption Threshold

We start with the simplest model, ignoring any internal structure of the film (membrane), which is represented just by a potential near the film (located at the plane x=0):(4)$$U\left(x\right)=-Tu\varphi \left(\left|x\right|/\Delta \right)$$
where U(x) is the potential energy *per unit length* of the chain, *T* is the temperature in energy units $(T=k_{B}T_{abs}$ with $T_{abs},$ the absolute temperature), *u* is the energy parameter, Δ is microscopic length defining the potential range, and φ is a non-dimensional function specifying a fast decay of the potential with the distance $\left|x\right|$ from the film. In the simplest case, the potential shape is rectangular: φ=1 for $\left|x\right|<\Delta$, and φ=0 otherwise.

It is always assumed that the chain is sufficiently rigid, l≫Δ, and that its contour length *L* is large, L≫λ, where the microscopic length-scale λ is defined below (cf. Equation (Equation 7)). In this case, the wormlike chain gets strongly trapped near the film for $u>u^{*}$, where $u^{*}$ is the adsorption threshold, which is defined mainly by *l* and Δ (and, strictly speaking, also by the function φ(x)). It can be estimated as
(5)$$u^{*}∼1/{\left(\Delta^{2}l\right)}^{1/3}$$

A similar estimate was obtained for the case of adsorption to an impenetrable surface [18,21,27]. The result, Equation (Equation 5), can be deduced from two statements: If $u\gg u^{*}$ the chain must be localized near the adsorbing plane. By contrast, for $u\ll u^{*}$ the chain statistics are just weakly affected by the adsorbing potential (provided that the chain is not too long). Both statements are justified below: Let us consider a chain fragment with one end located at x=0 and oriented parallel to the plane. By virtue of the wormlike flexibility implied by the Kratky–Porod chain model [24] adopted here (note that most of semirigid polymers are well-described by this model), the tangential vector $\underline{n}=\underline{n}\left(s\right)$ gradually changes along the chain contour (here *s* is the curvilinear coordinate). The typical angle between $\underline{n}\left(s\right)$ and the adsorbing plane is $\theta \left(s\right)∼{\left(s/l\right)}^{1/2}$ for s≪l [18,21,22]. (Note that θ is also the typical angle between $\underline{n}\left(0\right)$ and $\underline{n}\left(s\right)$ as follows from Equation (Equation 2)). The corresponding typical deviation along the *x*-axis is
(6)$$x\left(s\right)∼s\theta \left(s\right)∼s^{3/2}/l^{1/2}$$

Once x(s) exceeds the attraction length Δ, a correction to the chain trajectory is required in order to keep the chain in the Δ-layer, which costs about the thermal energy *T*. The corresponding fragment length is s∼λ, where
(7)$$\lambda ∼l^{1/3}\Delta^{2/3}$$
is the relevant microscopic length-scale along the chain. The whole chain confinement free energy (entropic cost) is, therefore, $F_{conf}∼TL/\lambda$, while its attraction (potential) energy is E∼−TLu. Therefore, the total adsorption free energy $F=F_{conf}+E$ is negative for $u\gg u^{*}∼1/\lambda$, and hence the chain confinement in the Δ-layer (i.e., in the region −Δ<x<Δ where polymer/membrane attraction is significant) is favorable in this case.

To justify the second statement, it is sufficient to estimate the upper bound for the total interaction energy, assuming that one chain end is pinned in the Δ-layer being oriented tangentially to it. The probability that a point *s* of the chain (s≳λ) is inside the Δ-layer is p(s)∼Δ/x(s). On using Equations (Equation 6) and (Equation 7) for λ≲s≲l we therefore obtain
(8)$$p\left(s\right)∼l^{1/2}\Delta /s^{3/2}∼{\left(\lambda /s\right)}^{3/2}$$

Obviously p(s)∼1 for s≲λ. For s≳l the chain is nearly Gaussian, so $x\left(s\right)∼l{\left(s/l\right)}^{1/2}$ and $p\left(s\right)∼\Delta /{\left(sl\right)}^{1/2}$. The total interaction energy is therefore
$$E∼-uT\int_{0}^{L}p\left(s\right)ds∼-uT\left(\lambda +\sqrt{L/l}\Delta \right)$$

The chain interactions with the adsorbing membrane can be neglected if E/T≪1; this condition is valid (in the regime $u\ll u^{*}$) provided that
(9)$$L≲l{\left(l/\Delta \right)}^{2/3}\equiv L_{m}$$

Note that $L_{m}\gg l$. In the next subsections, we consider the adsorption of relatively short, stiff, wormlike polymers with L≲l; both Equations (Equation 5) and (Equation 9) are valid in this case. Moreover, as discussed in Section 4, Equation (Equation 5) is generally valid, provided that $u^{*}$ is properly interpreted as the critical point for strongly anisotropic adsorption.

### 2.2. Adsorbed Layer Structure and Concentration Profile

Below, we build a scaling picture of a polymer layer critically adsorbed to a penetrable film for the case λ≪L≲l. Longer chains (L≫l) are considered in Section 3, Section 4 and Section 6. Generally, exactly at the adsorption threshold $u=u^{*}$, the chain is not yet trapped by the adsorbing surface (since the net free energy gain per segment is close to zero at $u=u^{*}$, see discussion in Section 7), so it is useful to demand that both ends of the chain are located in the attraction Δ-layer and are oriented parallel to it (to avoid their escape into the bulk of the solution). In this case, the layer can be viewed as a system of trains and loops: the trains are fragments of length $s_{train}≳\lambda$ located entirely in the Δ-layer, and the loops are the fragments between the trains (see Figure 1a). Note that the typical angle between any train segment and the membrane plane is
(10)$$\theta_{\Delta }={\left(\lambda /l\right)}^{1/2}∼{\left(\Delta /l\right)}^{1/3}$$

It is important that loops can intersect the Δ-layer, but each intersecting fragment (labeled as ‘int’ in Figure 1a) is short: its length $s_{int}\ll \lambda$ (otherwise, this fragment would be considered as a train), so the energy of each intersection $E_{int}$ is small, $\left|E_{int}\right|\ll T$. It is, therefore, plausible to assume that interactions with the Δ-layer just weakly perturb the statistics of a loop, which remains (nearly) ideal. To prove that this assumption is self-consistent, let us estimate the total interaction energy $E_{int}^{tot}$ of a single loop of length *s*. Reiterating the argument given at the end of Section 2.1, we obtain (cf. Equation (Equation 8))
$$E_{int}^{tot}\left(s\right)∼-2uT\int_{0}^{s/2}p\left(s\right)ds∼T$$

The partition function of a loop, $Z_{l}\left(s\right)$, s≫λ, must be therefore of the same order as that of an ideal loop whose segments do not interact with the film. The latter can be estimated as just the probability P(s) that an *s*-fragment starting in Δ-layer with initial orientation $\theta \left(0\right)∼\theta_{\Delta }$ has its end-point in the same layer with a similar orientation: x∼Δ, $\theta ∼\theta_{\Delta }$ at the end point. Recalling that the typical unconstrained values are much larger, x(s)≫Δ, $\theta \left(s\right)\gg \theta_{\Delta }$, for s≫λ (cf. Equation (Equation 6)), we obtain $P\left(s\right)∼\theta_{\Delta }\Delta /\left[x(s)\theta (s)\right]$.

Therefore,
(11)$$Z_{l}\left(s\right)∼P\left(s\right)∼\frac{l^{1/2}\Delta }{s^{3/2}}\frac{\lambda^{1/2}}{s^{1/2}}∼\frac{\lambda^{2}}{s^{2}}$$

The above scaling law is applicable to loops with s≫λ, while shorter loops with s∼λ are similar to trains. We therefore arrive at a self-similar distribution of loops: the number density of loops of size *s* is proportional to $Z_{l}\left(s\right)$. The concentration profile can be obtained using Equation (Equation 11) and the following argument was already employed in refs. [21,23]: The polymer concentration at the distance x≫Δ from the membrane is defined by loops of such length *s* that x(s)∼x, i.e., $s∼\lambda {\left(x/\Delta \right)}^{2/3}$. The number of monomers in such loops is $∼\int_{s/2}^{s}s^{\prime }Z_{l}\left(s^{\prime }\right)ds^{\prime }∼s^{2}Z_{l}\left(s\right)$ which must be proportional to $\int_{x/2}^{x}c\left(x^{\prime }\right)dx^{\prime }∼xc\left(x\right)$. As $s^{2}Z_{l}\left(s\right)=const,\;$ we obtain
(12)$$c\left(x\right)\propto 1/\left|x\right|$$
so that the relevant critical exponent is β=1; it is different from the result for an impenetrable film, β=4/3 [21]. (Note that the polymer actually forms two equivalent and connected adsorbed layers on both sides of the film). Equation (Equation 12) is valid for $\Delta \ll \left|x\right|\ll x_{max}$, where
(13)$$x_{max}=x\left(L\right)∼L^{3/2}/l^{1/2}$$

Equation (Equation 12) implies that most of polymer segments are far from the film (i.e., they belong to loops). The fraction *p* of segments in the Δ-layer is
(14)$$p∼c\left(\Delta \right)\Delta /\int_{0}^{x_{max}}c\left(x\right)dx$$
leading to p∼1/ln(N), where N≡L/λ. The total potential energy of critically adsorbed chain (at $u=u^{*}$) is
(15)E=−uTLp∼−TN/lnN
and the number of polymer contacts with the film is $n_{c}\propto \tilde{N}\equiv N/\ln\left(N\right)$ which formally corresponds to ϕ=1 (cf. Equation (Equation 1)).

### 2.3. Dependencies of Energy, Free Energy and Concentration Profiles on the Attraction Strength u

In this section, we continue to consider the regime λ≪L≲l for a chain with ends located in the Δ-layer. At u=0, there is no polymer/film interactions, and the chain behaves as a single loop. In this case, the monomer concentration at x∼x(s) is $c_{0}\left(x\right)∼s/x\left(s\right)$ leading to the power law: $c_{0}\left(x\right)\propto x^{-1/3}$ for $\left|x\right|≲x_{max}$ at u=0 (non-adsorbed state). By contrast, $c_{1}\left(x\right)\propto x^{-\beta }$ for $u=u^{*}$ (critically adsorbed state). In the general case, for $u<u^{*}$, the concentration profile can be represented (with the scaling accuracy) by $c_{1}\left(x\right)$ for short *x*, x<Λ, and $c_{0}\left(x\right)$ for x>Λ (cf. ref. [21]), leading to the following interpolation:(16)$$c\left(x\right)\propto {\left(x/\Lambda \right)}^{-1/3}+{\left(x/\Lambda \right)}^{-\beta },\;\;u<u^{*}$$
where Λ depends on the deviation τ from the critical point, $\tau \equiv u/u^{*}-1$. At τ=0, the profile is self-similar (cf. Equation (Equation 12)), so the first term in Equation (Equation 16) can be neglected. It is important, however, that the critically adsorbed state (corresponding to the second term in Equation (Equation 16)) is highly sensitive to a variation of τ. Indeed,
(17)$$\frac{\partial F}{\partial \tau }=-u^{*}TLp∼-T\tilde{N}\;\mathrm{at}\;\tau =0$$
where *F* is the free energy of the adsorbed chain, $E_{\Gamma }$ is the potential energy of a configuration Γ, $E_{\Gamma }=-uTLp_{\Gamma }$ (cf. Equation (Equation 15)), $\tilde{N}=N/\ln N$, and we used the theorem on small variations [28]:(18)$$\frac{\partial F}{\partial \tau }=\langle \frac{\partial E_{\Gamma }}{\partial \tau }\rangle =-u^{*}TL\langle p_{\Gamma }\rangle$$

Here $\langle ⊙\rangle$ meaning the ensemble average of ⊙ over all configurations and $\langle p_{\Gamma }\rangle =p$. Therefore,
$$\Delta F\equiv F\left(\tau \right)-F\left(0\right)≃-T\tilde{N}\tau$$
for a small $\left|\tau \right|$, so that a small negative τ can significantly destabilize the self-similar state (cf. Equation (Equation 12)): for τ<0, $\left|\tau \right|\gg 1/\tilde{N}$, its free energy increases by ΔF≫T, and its statistical weight decreases by a large factor $e^{\Delta F/T}\gg 1$, leading to $\Lambda \ll x_{max}$ (cf. Equation (Equation 16)). On the other hand, at $\left|\tau \right|∼1/\tilde{N}$, the destabilization is marginal and $\Lambda ∼x_{max}$, i.e.,
$$\Lambda ∼x_{max}∼N^{3/2}\Delta \;\;\mathrm{for}\tau =\tau^{*}=-\ln N/N$$

Assuming that Λ is a function of τ only (i.e., it is nearly independent of *N*) as long as $\left|\tau \right|≳1/\tilde{N}$ (which is equivalent to $x_{max}≳\Lambda$) and neglecting the lnN factor, we find
(19)$$\Lambda ∼\tau^{-3/2}\Delta ,\;\;\left|\tau \right|\gg 1/N,\;\tau <0$$

The τ-dependence of the total energy is defined by equation $E=-u^{*}TL\left(1+\tau \right)p$. Using Equations (Equation 16), (Equation 19) and (Equation 14), we thus obtain for τ<0, $1/N\ll \left|\tau \right|\ll 1$:(20)$$p∼{\left(\left|\tau \right|N\right)}^{-1}$$
(21)$$E∼-T/\left|\tau \right|$$

The dependence of the chain free energy *F* on τ can be deduced from Equation (Equation 18):(22)$$\frac{\partial F}{\partial \tau }=-u^{*}TLp$$ (cf. Equation (Equation 17)). Using also Equation (Equation 20) we get for τ<0:(23)$$F\left(\tau \right)-F\left(0\right)∼T\ln\left(\left|\tau \right|N\right),\;\;1/N\ll \left|\tau \right|\ll 1$$

Note that for u=0(τ=−1), the chain partition function is $Z_{l}\left(L\right)\propto \frac{1}{x(L)\theta (L)}\propto 1/N^{2}$, so
(24)F(−1)≃2TlnN
suggesting (by virtue of Equation (Equation 23)) that F(0)∼TlnN. (Note that by virtue of Equations (Equation 14) and (Equation 16), the assumption adopted above Equation (Equation 19) is equivalent to assuming that the number of polymer/membrane contacts $n_{c}\left(\tau \right)\propto pN$ depends on τ but not on the chain length at τ<0, $\left|\tau \right|n_{c}\left(0\right)\gg 1$. This is a standard assumption of the scaling theory of polymer adsorption [4]).

Let us turn to the regime of adsorbed chain, $u>u^{*}$, i.e., τ>0. The reduced adsorption energy per unit length,
ϵ≃−F/(TL)
gets positive and significant in this regime (in contrast to the pre-adsorption regime, τ<0, where Equations (Equation 23) and (Equation 24) lead to $\left|\epsilon \right|∼1/L$). The dependence of ϵ on τ for τ>0 can be obtained using the general Equation (Equation 22):(25)$$\frac{\partial \epsilon }{\partial \tau }=u^{*}p$$

To obtain *p* using Equation (Equation 14), we need to know the concentration profile for τ>0. To this end, we can use the same argument as that proposed in Section 2.2 to obtain the self-similar profile at τ=0. The only difference is that for τ>0 (and hence, as verified below, ϵ>0) the partition function $Z_{l}\left(s\right)$ of a loop (with s≫λ) must be replaced by a modified one:(26)$$Z_{l\epsilon }\left(s\right)=Z_{l}\left(s\right)e^{-\epsilon s}\propto s^{-2}e^{-\epsilon s}$$
where $Z_{l}\left(s\right)$ is defined in Equation (Equation 11). Here the statistical factor $e^{-\epsilon s}=e^{-\Delta F/T}$ accounts for the free energy penalty ΔF=Tsϵ for the desorption of a large *s*-loop we consider. For $s<s^{*}=1/\epsilon$, the new factor is not important, so the critical law, Equation (Equation 12), stays valid for x≪h:(27)$$c\left(x\right)\propto x^{-1},\;x\ll h$$
where
(28)$$h\equiv x\left(s^{*}\right)=l^{-1/2}\epsilon^{-3/2}$$ (here we also assume that $h\ll x_{max}$, that is ϵL≫1). Obviously, for x>h$(s>s^{*}$), the loops are suppressed, so the concentration c(x) decays exponentially. The decay law for x≫h was established in ref. [21]:(29)$$c\left(x\right)\propto \exp\left(-\frac{4}{3}{\left(\frac{12x}{h}\right)}^{1/2}\right),\;x\gg h$$

It is clear that the region x≫h can be neglected in the integral of Equation (Equation 14), which gives
(30)$$p∼1/\ln\left(h/\Delta \right)=-k_{1}/\ln\left(\epsilon \lambda \right)$$
where $k_{1}$ is a numerical factor. Equations (Equation 25) and (Equation 30) then give
$$\lambda \frac{d\epsilon }{d\tau }=-k_{2}/\ln\left(\epsilon \lambda \right)$$
where $k_{2}=k_{1}u^{*}\lambda$ is another numerical factor (cf. Equations (Equation 5) and (Equation 7)). The above equation leads to
(31)$$\epsilon ≃\left(k_{2}/\lambda \right)\tau /\ln\left(1/\tau \right),\;1\gg \tau \gg 1/N$$

The obtained critical behavior of the free energy per unit length points to a second-order phase transition at τ=0. According to Equation (Equation 28), the thickness of the adsorbed layer, *h*, rapidly decreases with τ roughly as $h\propto \tau^{-3/2}$. By contrast, the fraction *p* of monomers in contact with the film (in the Δ-layer) increases logarithmically, p∝1/ln(1/τ) for 1≫τ≫1/N.

## 3. Quantitative Approach

The scaling results described above can be also justified with a quantitative theory accounting for both spatial and orientational distributions of polymer segments. Following ref. [21], let us consider a semirigid chain of length *L* with one free end and the second end having position *x* and orientation $\underline{n}$ (cf. Figure 1b).

The partition function of such an ideal chain in the presence of the membrane potential (U(x) at $\left|x\right|<\Delta$) is $\psi (x,\underline{n},L)$. Its change as *L* grows can be described by the transfer-matrix approach, leading to the master equation (the generalized Edwards equation) [18,21,29,30]:(32)$$\frac{\partial \psi }{\partial L}=\frac{1}{l}\nabla_{\underline{n}}^{2}\psi -n_{x}\frac{\partial \psi }{\partial x}-\frac{U(x)}{T}\psi$$ (note that ψ does not depend on the y,z coordinates in the membrane plane). At large *L*, the function ψ must tend to the ground state with
$$\psi \left(x,\underline{n},L\right)\to e^{\epsilon L}\psi \left(x,\underline{n}\right),$$
where $\psi (x,\underline{n})$ is the ground eigenfunction (corresponding to the largest ϵ) of the operator on the rhs of Equation (Equation 32):(33)$$\frac{1}{l}\nabla_{\underline{n}}^{2}\psi -n_{x}\frac{\partial \psi }{\partial x}-\frac{U(x)}{T}\psi =\epsilon \psi$$

A similar equation was derived in ref. [21]. Here
(34)$$\epsilon \equiv \lim_{L\to \infty }\left(-F/(TL)\right)$$
where *F* is the chain free energy increment due to interactions with the membrane. Below, we focus on the layer structure at distances $\left|x\right|\ll l$ in the limit of long chains L≫l. Taking into account that $\underline{n}$-dependence of ψ reduces to its dependence on the angle $\theta ≃n_{x}$ between $\underline{n}$ and the plane, and that $\left|\theta \right|\ll 1$ for $\left|x\right|\ll l$ (the condition $\left|\theta \right|\ll 1$ is suggested by the scaling analysis in Section 2.2), we rewrite Equation (Equation 33) as (in analogy with ref. [21])
(35)$$\frac{1}{l}\frac{\partial^{2}\psi }{\partial \theta^{2}}-\theta \frac{\partial \psi }{\partial x}-\frac{U(x)}{T}\psi =\epsilon \psi$$
where ψ=ψ(x,θ). This equation is obviously invariant with respect to substitutions x→−x, θ→−θ, implying an obvious symmetry
(36)ψ(−x,−θ)=ψ(x,θ)

Furthermore, as the film is penetrable, we should demand that
(37)$$\psi \left(-\Delta ,\theta \right)=\psi \left(\Delta ,\theta \right)\;\mathrm{for}\theta \gg \theta_{\Delta }\equiv {\left(\Delta /l\right)}^{1/3}$$

The above equation simply states that partition functions of chains with ends at $A_{-}$ and $A_{+}$ (cf. Figure 2) are nearly equal. The condition $\theta \gg \theta_{\Delta }$ provides that (i) the potential energy due to the end segment (of length $s_{int}≃2\Delta /\theta$) of chain $A_{+}$ penetrating the membrane is low (as compared to *T*), so that the potential energies of the two chains are nearly equal, and (ii) the bending of the $s_{int}$-segment is negligible compared to θ so that the end angles of both chains are nearly equal to θ (cf. Figure 2).

Our primary goal is to find ψ(x,θ) at the critical point $u=u^{*}$. As discussed in Section 2.3 at this point ϵ vanishes (actually, it nearly vanishes and can be neglected, see Section 4 below). It was also shown in Section 2.2 that the concentration profile at $u=u^{*}$ follows an algebraic law, Equation (Equation 12), for
(38)$$l\gg \left|x\right|\gg \Delta$$

It is therefore reasonable to assume (again following ref. [21]) a self-similar *x*-dependence of ψ (in the regime of Equation (Equation 38)):(39)$$\psi \left(x,\theta \right)=x^{\alpha }f\left(\theta /\theta_{x}\right),\;x>0$$
where α is a constant to be determined, and $\theta_{x}$ is a function of *x*. To obtain ψ(x,θ) for x<0, one should use the symmetry relation, Equation (Equation 36). The dependence $\theta_{x}={\left(x/l\right)}^{1/3}$ simply follows from the structure of Equations (Equation 35) and (Equation 39) with ϵ=0 and U(x)=0 for x>Δ; it was obtained in ref. [21]. As a result, we arrive at the following equation for f(η) with $\eta \equiv \theta /\theta_{x}$ (cf. ref. [21]):(40)$${\frac{\partial^{2}f}{\partial \eta^{2}}}_{}+\frac{1}{3}\eta^{2}\frac{\partial f}{\partial \eta }-\alpha \eta f=0$$ (Note that η here has the opposite sign compared to ref. [21]). The boundary condition for f(η) stemming from Equation (Equation 37) is:(41)f(−η)≃f(η)forη→∞

It is crucially different from that adopted in ref. [21]. Following the approach of ref. [21], the differential Equation (Equation 40) can be reduced to the Kummer equation, and f(η) can be expressed in terms of degenerate hypergeometric functions:(42)$$f\left(\eta \right)=\Psi \left(-\alpha ,2/3,-\eta^{3}/9\right)$$
which is selected by the condition that f(η) must not increase exponentially at $\left|\eta \right|\to \infty$. Here
(43)$$\Psi \left(a,b,z\right)=\Gamma \left[\begin{array}{c} 1-b\\ a-b+1 \end{array}\right]F\left(a,b,z\right)+\Gamma \left[\begin{array}{c} b-1\\ a \end{array}\right]z^{1-b}F\left(a+1-b,2-b,z\right)$$
is the Tricomi function, and F(a,b,z) is the standard degenerate hypergeometric function ${(}_{1}F_{1}$). For −η≫1 the function, Equation (Equation 42), shows a power law behavior:(44)$$f\left(\eta \right)≃3^{2\alpha }{\left(-\eta \right)}^{-3\alpha }$$

The parameter α can be obtained based on the physical boundary condition, Equation (Equation 41), which leads to the following equation:(45)$$\sin\left(\frac{2\pi }{3}\right)+\sin\left(\pi \alpha \right)=\sin\left(\frac{2\pi }{3}+\pi \alpha \right)$$

It has two series of roots:α=2n,α=2n−2/3
with integer *n*. As we look for ground states, f(η) must not have any knots at finite η. This condition selects just two values: $\alpha =\alpha_{0}=0$ and $\alpha =\alpha_{1}=-2/3$. The corresponding eigenfunctions are
(46)$$\psi_{0}\left(x,\theta \right)=1,\;\;\psi_{1}\left(x,\theta \right)=x^{-2/3}\Psi \left(2/3,2/3,-\theta^{3}l/\left(9x\right)\right)$$

For $\left|\theta \right|\gg {\left(x/l\right)}^{1/3}$, we obtain (cf. Equation (Equation 44)) $\psi_{1}\left(x,\theta \right)≃const\;\theta^{-2}$. Obviously, $\psi_{0}$ reflects an isotropic state at u=0, while $\psi_{1}$ describes the strongly anisotropic critically adsorbed state at $u=u^{*}$. In other words, $\psi_{0}$ corresponds to the trivial isotropic adsorption point (u=0) which is characteristic for both ideal flexible and long semiflexible chains (see the next section for a further discussion).

The monomer density distribution in the (x,θ) space is [21,30,31]
(47)c(x,θ)=constψ(x,θ)ψ(x,−θ)

Therefore, $c_{0}\left(x,\theta \right)=const\;$, i.e., the chain stays free in the pre-adsorbed state (u=0), its monomers are distributed uniformly and isotropically. By contrast, the distribution $c_{1}\left(x,\theta \right)$ in the critical state for strong adsorption (corresponding to $u=u^{*},\;\alpha =\alpha_{1}$) is highly anisotropic and *x*-dependent:(48)$$c_{1}\left(x,\theta \right)=C_{1}\;{\left(x/l\right)}^{-4/3}\rho \left(\theta /\theta_{x}\right)$$
where $C_{1}=const$ and
(49)$$\rho \left(\eta \right)=C_{N}\Psi \left(2/3,2/3,-\eta^{3}/9\right)\Psi \left(2/3,2/3,\eta^{3}/9\right)$$

Here, $C_{N}\approx 0.044175$ is the normalization constant defined by the condition
(50)∫ρ(η)dη=1

The universal function ρ(η) defines the orientational distribution of all polymer segments at a distance *x*(Δ≪x≪l) from a penetrable membrane. This function is plotted in Figure 3.

Note that ρ(η) shows a power-law decay, $\rho \propto \eta^{-4}$, at large η$(\left|\eta \right|\gg 1$) as demonstrated in the inset of Figure 3. This is in contrast to a very strong compressed exponential decay predicted in ref. [21] for the case of adsorption to an impenetrable solid surface. Interestingly, for $\left|\theta \right|\gg \theta_{x}$, the $c_{1}\left(x,\theta \right)$ distribution depends on the angle θ, but is almost independent of *x*
(51)$$c_{1}\left(x,\theta \right)=const\;\theta^{-4},\;\;\left|\theta \right|\gg \theta_{x}$$

The reason for this behavior is simple: the relevant chain segments of length x/θ can be considered straight.

The monomer density as a function of distance only is c(x)=∫c(x,θ)dθ (here the integration limits are −π/2 and π/2, but note that for x≪l the integral mainly comes from $\left|\theta \right|∼\theta_{x}\ll 1$). At the adsorption threshold, we obtain $c_{1}\left(x\right)\propto x^{-1}$ for x≪l in agreement with the scaling result, Equation (Equation 12).

## 4. Dependence of the Adsorbed Layer Structure on the Attraction Strength

In the previous section, we rederived the critical exponent for the self-similar concentration profile at $u=u^{*}$ using the transfer-matrix theory. We are now in a position to consider the dependence of the adsorption profile on $\tau =u/u^{*}-1$. At this point, it is important to note that, strictly speaking, $u=u^{*}$ does not correspond to the chain-trapping transition: a purely attractive potential U(x) we consider always traps a long (L→∞) chain for whichever low attraction strength *u*. This statement is well-known for flexible chains [4], and it is also valid for wormlike chains, which behave as flexibly at length scales ≫l [21]. To show this, let us consider a long ideal wormlike chain (L≫l) near an attractive penetrable membrane with $0<u<u^{*}$. Assuming that the chain conformation is unperturbed by the potential, one can easily estimate the total potential energy of such a chain:$$E∼-uTL\Delta /R∼-uT\Delta \sqrt{L/l}$$
where *R* is the chain size defined in Equation (Equation 3). The chain is trapped by the membrane if $\left|E\right|\gg T$, i.e., for
$$u\gg \Delta^{-1}\sqrt{l/L}$$

Hence, for L→∞, the chain is indeed trapped for any u>0. This conclusion stays valid also if a perturbation of the chain conformation by the membrane potential is taken into account since such a perturbation must always increase $\left|E\right|$.

Then the natural question arises: what is the meaning of $u^{*}$ if the adsorption occurs already at $u=0^{+}$? The answer is that $u^{*}$ defines the second transition to a strongly anisotropic adsorbed state accompanied by the formation of a self-similar layer. This point and the features of both transitions are further discussed below.

The argument given above shows that the reduced adsorption energy per unit length, ϵ, is always positive for u>0. Let us consider the distal region, x≫l, of the adsorbed layer. Here we can apply the classical equation (cf. ref. [32] and Equation (Equation 25) of ref. [21]) for an ideal flexible chain:(52)$$\frac{l}{6}\frac{\partial^{2}\psi }{\partial x^{2}}=\epsilon \psi$$
where ψ=ψ(x) is the partition function of the chain with one end at *x*, and ϵ>0 is considered as known parameter. The solution to the above equation for the concentration profile $c\left(x\right)\propto \psi {\left(x\right)}^{2}$ is
(53)$$c\left(x\right)≃c_{0}\;\exp\left(-\left|x\right|/h\right)$$
where
(54)$$h=\frac{1}{2\sqrt{6}}\sqrt{l/\epsilon }$$
is the distal layer thickness. To find *h*, we eventually need to establish the dependence of ϵ=ϵ(τ) on the attraction strength. To this end, we have to obtain the whole concentration profile c(x), and, in particular, its proximal part near the membrane, at x≲l. At $u\ll u^{*}$, nothing special happens with c(x) in this proximal region: Equation (Equation 53) remains valid there, and the same is true for $u∼u^{*}/2$. To see this, recall that the interaction energy of an *l*-segment intersecting the membrane at an angle θ is $E_{int}≃-2uT\Delta /\sin\theta =-E_{1}/\sin\theta$ (cf. Figure 2), where $E_{1}/T=2u\Delta ≲u^{*}\Delta ∼{\left(\Delta /l\right)}^{1/3}\ll 1$. Therefore, for most θ, the interaction energy is small $(\left|E_{int}\right|\ll T$); hence, the segment positional and orientational distribution is only slightly perturbed by the membrane potential. It remains nearly uniform and isotropic at x≲l in accordance with Equation (Equation 53). It is only at the lowest possible angles, $\theta ∼\theta_{\Delta }\ll 1$ (cf. Equation (Equation 37)), that the perturbation is moderate $(\left|E_{int}\right|∼T$), so that tangential orientations (with low θ) are somewhat preferred. Still the overall orientational order parameter $S_{or}=0.5\langle 3\sin^{2}\theta -1\rangle$ remains small: $\left|S_{or}\right|∼\theta_{\Delta }∼{\left(\Delta /l\right)}^{1/3}\ll 1$ (a logarithmic factor is neglected here).

The situation changes only at $u\approx u^{*}$$(\left|\tau \right|\ll 1$), where ‘trains’ (sequences of tangentially oriented segments) and loops start to form near the membrane. At $u=u^{*}$(τ=0), they result in a self-similar concentration profile, $c\left(x\right)\propto 1/\left|x\right|$, and strong ‘disklike’ orientation of polymer segments, $S_{or}≃-0.5$, in the proximal layer at $\left|x\right|\ll l$ (cf. Section 2.2 and Section 3). This structure is due to a fractal distribution of polymer loops. At a weaker attraction, $u<u^{*}$(τ<0), the self-similar region becomes thinner. To quantify this effect, let us firstly recall that the fractal structure of loops emerges at $u=u^{*}$ because in this case, a detachment of a ‘train’ (its conversion into a loop) never costs any significant free energy: the entropic free energy due to the ‘train’ confinement in the Δ-layer is exactly compensated by the negative energy gain due to the membrane potential. This balance is broken at τ<0 when the potential energy gain is decreased. As a result, a conversion of a ‘train’ of length s=gλ into a loop now leads to an additional negative free energy $\Delta F=(u-u^{*})sT∼\tau gT$, τ<0 (cf. Equations (Equation 5) and (Equation 7)). Its effect can be neglected for $g≲g^{*}∼1/\left|\tau \right|$ (i.e., $s\ll s^{*}∼\lambda /\left|\tau \right|$), but it destroys the self-similar structure at larger *s*. The thickness Λ of the fractal layer at τ<0 is, therefore, defined by the *x*-size of the crossover loops of length $∼s^{*}$:(55)$$\Lambda ∼\Delta /{\left|\tau \right|}^{3/2},\;\;\tau <0,\;\left|\tau \right|≳\lambda /l$$

Note that the above equation agrees with Equation (Equation 19). The condition $\left|\tau \right|≳\lambda /l$ reflects the fact that the maximal thickness of the self-similar layer is ∼l; hence Λ∼l for $\left|\tau \right|≲\lambda /l∼{\left(\Delta /l\right)}^{2/3}$. The whole adsorbed layer (at x>0) is shown schematically in Figure 4.

Thus, the self-similar structure of loops is predicted in the proximal sublayer, Δ<x<Λ. At the crossover, x∼Λ, the proximal loops (of length $s≲s^{*}$) readily conjugate with nearly isotropic larger loops of size ≳l, which dominate further away from the membrane. The conjugation means that concentration $c_{ll}\left(x,\theta \right)$ of due to large loops and $c_{ss}\left(x,\theta \right)$ due to self-similar loops (of size $s∼s^{*}$) must be comparable at x=Λ and $\theta ∼\theta_{\Lambda }∼{\left(\Lambda /l\right)}^{1/3}$ (recall that proximal loops are mostly oriented tangentially to the surface, cf. Equation (Equation 39)). Obviously $c_{ll}\left(x,\theta \right)∼c_{0}$ (cf. Equation (Equation 53)). As for $c_{ss}(x,\theta$), it is given in Equation (Equation 48): $c_{ss}\left(x,\theta \right)∼C_{1}\;{\left(x/l\right)}^{-4/3}$ for $\theta ≲\theta_{x}$. The conjugation condition therefore gives $C_{1}∼c_{0}{\left(\Lambda /l\right)}^{4/3}$ so that the total concentration $c≃c_{ll}+c_{ss}$ is
(56)$$c\left(x,\theta \right)∼c_{0}\left[1+{\left(x/\Lambda \right)}^{-4/3}\rho \left(\theta /\theta_{x}\right)\right],\;\Delta <x≲l$$

The total concentration profile obtained by integration over θ therefore is (note that $c_{ss}\left(x\right)=\int c_{ss}\left(x,\theta \right)d\theta ∼C_{1}l/x$)
(57)$$c\left(x\right)≃c_{0}\left[1+\frac{\tilde{x}}{x}\right],\;\;\Delta <x≲l$$
where
(58)$$\tilde{x}∼\frac{\Lambda^{4/3}}{l^{1/3}}$$ (Note that by continuity $c\left(x\right)∼c_{0}(1+\tilde{x}/\Delta$) for $\left|x\right|<\Delta$). The whole concentration profile at x>Δ is defined by Equations (Equation 53) and (Equation 57). The whole adsorbed layer can be viewed as a superposition of ‘trains’ at $\left|x\right|∼\Delta$, aligned self-similar loops and nearly isotropic longer fragments of size ≳l. The aligned loops disappear for τ<0, $\left|\tau \right|∼1$: in this case Λ∼Δ, so the loops are short and can be considered as parts of ‘trains’. Furthermore, the contribution of both aligned loops and trapped ‘trains’ can be neglected in the total concentration c(x) if $\left|\tau \right|≳{\left(\Delta /l\right)}^{1/6}$ (corresponding to $\tilde{x}≲\Delta$, cf. Equations (Equation 58) and (Equation 55)).

We are now in a position to obtain the dependence of ϵ and *h* on τ. To this end, we use the general Equation (Equation 25): $\frac{\partial \epsilon }{\partial \tau }=u^{*}p∼p/\lambda$, where *p* is the fraction of trapped monomers (cf. Equation (Equation 14))
(59)$$p=\int_{0}^{\Delta }c\left(x\right)dx/\int_{0}^{\infty }c\left(x\right)dx$$

Using c(x) defined above and Equations (Equation 55) and (Equation 58), we obtain (generally neglecting log-factors)
$$p∼\frac{\Delta +\tilde{x}}{h},\;\;$$
$$\tilde{x}∼\left\{\begin{array}{c} \frac{\Delta^{4/3}}{l^{1/3}}{\left|\tau \right|}^{-2},\;{\left(\Delta /l\right)}^{1/6}≳\left|\tau \right|≳{\left(\Delta /l\right)}^{2/3}\\ l,\;\;\;\;\;\;\left|\tau \right|≲{\left(\Delta /l\right)}^{2/3}\; \end{array}\right.$$

Solving Equation (Equation 25) with initial condition, ϵ=0 for $u/u^{*}=1+\tau =0$, we obtain for τ<0:(60)$$\epsilon l∼\left\{\begin{array}{ccc} \delta^{2}{\left(1+\tau \right)}^{2} & , & \left|\tau \right|≳\delta \\ {\left|\tau \right|}^{-2}\delta^{4} & , & \delta ≳\left|\tau \right|≳\delta^{2}\\ 1 & , & \left|\tau \right|≲\delta^{2} \end{array}\right.$$
where $\delta \equiv {\left(\Delta /l\right)}^{1/3}$. Note that ϵL≳1 is required for adsorption and that for $u\ll u^{*}$ Equation (Equation 60) gives $\epsilon L∼\left(L/l\right)\delta^{2}{\left(u/u^{*}\right)}^{2}$. The latter equation shows that ϵL≪1 and, hence, the chain in not adsorbed (and not really affected by attraction to the membrane) for $L≲l/\delta^{2}=L_{m}$ in agreement with Equation (Equation 9). By contrast, for $L\gg L_{m}$ and $u∼u^{*}/2$, the chain must be well-adsorbed (trapped by the membrane), but the proximal anisotropic layer is not yet formed since for $\tau \equiv u/u^{*}-1∼-1/2$ the crossover distance Λ∼Δ and c(x) is always dominated by the isotropic contribution of long semiflexible loops.

The terminal thickness *h* of the adsorbed layer (cf. Equation (Equation 54)) for τ<0 therefore is
(61)$$h∼\left\{\begin{array}{ccc} l\delta^{-1}{\left(1+\tau \right)}^{-1}=l^{4/3}\Delta^{-1/3}u^{*}/u & , & \left|\tau \right|≳\delta \\ l\left|\tau \right|/\delta^{2} & , & \delta ≳\left|\tau \right|≳\delta^{2}\\ l & , & \left|\tau \right|≲\delta^{2} \end{array}\right.$$

Thus all the 3 length-scales, *h*, Λ and $\tilde{x}$ becomes similar to the Kuhn segment *l* near the critical point $u=u^{*}$$(\left|\tau \right|≲\delta^{2}$) for the second (anisotropic) adsorption transition. At this point ϵ∼1/l, so the effect of ϵ can be neglected in the proximal anisotropic layer of relatively short aligned loops (whose length s≪l).

Above $u^{*}$ (at τ>0), the adsorbed layer structure is characterized by a single length-scale *h* (apart from the ‘nano-scale’ Δ). The concentration profile, ϵ and *h* can be obtained in exactly the same way as discussed in Section 2.3 (cf. Equations (Equation 27), (Equation 29) and (Equation 31)):$$c\left(x\right)∼const\;x^{-1}\exp\left(-\frac{4}{3}{\left(\frac{12x}{h}\right)}^{1/2}\right),\;x\gg \Delta$$
(62)$$\epsilon l∼\left\{\begin{array}{ccc} 1 & , & 0<\tau ≲\delta^{2}\\ \delta^{-2}\tau /\ln\left(1/\tau \right) & , & 1≳\tau \gg \delta^{2} \end{array}\right.$$
(63)$$h≃l/{\left(\epsilon l\right)}^{3/2}∼\left\{\begin{array}{ccc} l & , & 0<\tau ≲\delta^{2}\\ \Delta {\left[\ln(1/\tau )/\tau \right]}^{3/2} & , & 1≳\tau \gg \delta^{2} \end{array}\right.$$

The layer thickness *h* therefore decreases with τ; it changes from h∼l at τ=0 to h∼Δ at τ∼1.

## 5. Preliminary Discussion

In the previous sections, we considered the adsorption of wormlike polymers to a flat penetrable membrane. Modeling a membrane (a free-standing film) as just a potential energy profile, which is uniform along the membrane plane (cf. Equation (Equation 4)), may seem to be far from reality. However, in the spirit of a mean-field approach, it is really plausible to represent a film by an effective potential of its interaction with the adsorbed macromolecule. The mean molecular potential can reflect the film structure in the normal direction (across the film). On the other hand, the heterogeneous film structure in the lateral direction can be neglected if the corresponding correlation length is smaller than the characteristic lateral size λ associated with the wormlike polymer (cf. Equation (Equation 7)).

Nevertheless, it would be useful to consider some specific physical realizations of penetrable membranes. It is well-known that many polymers tend to adsorb to surfactant layers, including free-standing double-layer surfactant vesicles and lipid bilayers [33]. Interfaces between surfactant layers and polymers are known to be highly relevant to biophysical membrane problems [34]. In particular, the adsorption of semiflexible polymers at the exterior side of a membrane can induce changes of a vesicle or a biological cell shape [8]. Interactions of macromolecules with cell membranes are important for many processes in cell biology [8,10,11,35,36] including DNA uptake by cells. Biopolymers such as DNA normally use appropriate pores to penetrate cell membranes [37,38]. Importantly, it was recently established that a direct penetration across the cell membrane is possible for amphipathic phospholipid polymers mimicking the chemical structure of phospholipids in the membrane [9]. It is therefore reasonable to expect that a surfactant (lipid) bilayer in a disordered liquid state can be penetrable by a locally stiff macromolecule if it has some affinity to non-polar hydrophobic molecular fragments (tails) inside the bilayer. The macromolecule must be therefore amphiphilic with alternating polar/charged groups (to ensure aqueous solubility) and hydrophobic fragments. On the other hand, the macromolecular adsorption can be due to electrostatic attraction of ionic groups to oppositely charged ‘heads’ of surfactants in the bilayer membrane. A high salinity of the solution can be used to keep the electrostatic interactions sufficiently short-range so that the simple model of Equation (Equation 4) may be applicable in this case.

One should bear in mind, however, that the very penetration of a polymer end into the bilayer may imply a transient increase in the total free energy due to a disruption of the layer structure in a correlation volume near the penetration location. The penetration would then require to overcome an activation energy barrier. This feature is totally irrelevant for the equilibrium microstructure of the adsorbed polymer layers considered in this paper, but it may significantly slow down the layer formation and equilibration.

Instead of a free-standing surfactant layer, one can consider adsorption onto a simple interface between two immiscible solvents, such as water and a hydrophobic oily liquid. Here, the role of an adsorbing macromolecule can be taken by a weakly charged polyelectrolyte with a stiff (semiflexible) backbone. (Recall that most semiflexible biopolymers, such as double-strand DNA or F-actin, are charged). Such polyelectrolyte molecules can be soluble in both phases (in water, due to charged groups, and in oil, due to a hydrophobic backbone). The driving force for adsorption of such wormlike polymers at the interface can be provided by their amphiphilic nature: both types of polymer groups can benefit from a favorable environment near the interface, for example, if the charged groups are grafted to the hydrophobic backbone with flexible spacers. The general problem here is that the polymer, being soluble in both phases, may still prefer one of them. It can be possible, however, to restore the symmetry (to render the solvation free energies in both solvents equal) by tuning the ion strength (or pH) in the water phase.

Yet another possibility is to consider an ultra-thin solid film with nano-holes in it. For the theoretical approach developed in this paper to be applicable, the nano-holes (of diameter *D*) must be much larger than the film attraction range Δ, but much smaller than the polymer rigidity segment *l*: Δ≪D≪l (in addition, $D\theta_{\Delta }$ must be larger than the chain thickness *d*). A polymer adsorption layer can be then naturally formed due to polymer/film attraction, while, in addition, the semiflexible polymer fragments can easily penetrate the layer through the holes. Modern technologies offer a number of techniques to create such porous ultra-thin membranes [39,40,41,42,43,44]. In particular, 1 nm thick carbon nano-membranes with pores of diameter 3–30 nm were produced [45].

In the next section, we show that complexes of wormlike polymers with such porous membranes show some universal features, which require a special analysis provided there.

## 6. Partially Penetrable Membranes

Let us consider an ultra-thin solid nanolayer (for example, a free-standing carbon, metal or polymer nanosheet) of effective thickness 2Δ (including its attraction zone, see Figure 5) perforated with nano-holes of diameter ∼D. The regime of interest is Δ≪D≪l. We keep considering a very long semiflexible chain, L≫l. The polymer fragments can penetrate the layer through the holes. Otherwise, the adsorbed fragments can stay near the solid surface, while polymer penetration inside the solid film is not allowed (cf. Figure 5). The membrane attraction potential is defined, as before, in Equation (Equation 4), where *x* should be now replaced by the distance to the solid surface (so, of course, the potential is virtually absent in the hole regions). The important parameter is the area fraction *w* of holes. The mean total area per hole is $∼D^{2}/w$. The mean distance between the centers of neighboring holes is therefore $∼D/\sqrt{w}$. To simply the argument, we also assume that this distance exceeds the microscopic length λ along the membrane, $D/\sqrt{w}≳\lambda$.

At the adsorption threshold, $u=u^{*}$, the polymer creates a structure of loops. By a loop, here we mean a chain fragment whose both ends are located near the membrane surface (in the Δ-layer next to it) and are oriented nearly parallel to it. Such a loop can intersect the x=0 plane. Let us focus on the loops with length *s* in the range $D/\sqrt{w}\ll s\ll l$. It corresponds to the following range of *x*-coordinates: $x_{min}\ll \left|x\right|\ll l$, where $x_{min}=x\left(D/\sqrt{w}\right)=D^{3/2}/\left(l^{1/2}w^{3/4}\right)$ (cf. Equation (Equation 6)). It must be expected that a self-similar structure of loops is formed in this regime since the heterogeneous structure of the film becomes effectively coarse-grained at $\left|x\right|\gg x\left(D/\sqrt{w}\right)$. As a result, each intersection of the nanolayer by the chain now brings an additional statistical weight factor *w* in the partition function (note that *w* is just the probability that the chain ‘finds’ a hole). Thus, the only difference with the reference model (Section 2) is related to the free energy cost of an intersection. Before (in Section 2, Section 3 and Section 4), it was equal to $E_{int}=-2uT\Delta /\sin\theta_{cross}$, now it is −Tlnw independent of the crossing angle $\theta_{cross}$.

Let us turn to the structure of loops for the membrane with holes. To start with, we switch off any interactions with the membrane (located around the x=0 plane). Let $Z_{l}\left(s|n\right)$ denote the partition function of a loop of contour length *s* involving *n* intersections with the plane, x=0. The function $Z_{l}\left(s|0\right)$ for n=0 was considered in ref. [21] (and denoted as $Z_{l}\left(s\right)$ there):$$Z_{l}\left(s|0\right)∼{\left(s/\lambda \right)}^{-5/2},\;s≲l$$

On the other hand, the total partition function of a loop (with arbitrary number of intersections) for a perfectly permeable adsorbing film is
(64)$$Z_{l}\left(s\right)=\sum_{n}Z_{l}\left(s|n\right)$$

As was established in Section 2.2 (Equation (Equation 11))
(65)$$Z_{l}\left(s\right)∼{\left(s/\lambda \right)}^{-2}$$

Therefore, the probability p(s|0) that a loop has no crossings is
(66)$$p\left(s|0\right)\equiv Z_{l}\left(s|0\right)/Z_{l}\left(s\right)∼{\left(s/\lambda \right)}^{-0.5}$$

It is not difficult to estimate the *mean* number of intersections of a loop of length *S*, $\bar{n}\left(S\right)$. With the same scaling accuracy (i.e., omitting the numerical prefactor), the result is
(67)$$\bar{n}\left(S\right)≃const\;\ln\left(S/\lambda \right)$$

The prefactor in the above equation can be obtained directly using the obvious relation $\bar{n}\left(S\right)≃2\int_{\lambda }^{S/2}P\left(s\right)ds$, where P(s)ds is the probability that a chain element (s,s+ds) intersects the x=0 plane (s is the contour length from the nearest end of the loop). Obviously $P\left(s\right)ds≃\langle \left|\theta (s)\right|\delta \left(x\left(s\right)\right)\rangle ds∼\langle \left|\theta (s)\right|\rangle ds/\langle \left|x(s)\right|\rangle$, where θ(s) is the tilt angle and x(s) is the *x*-coordinate of the point *s* of the loop (recall that the typical $\theta \left(s\right)∼{\left(s/\lambda \right)}^{0.5}$ and x(s)∼sθ, cf. Equation (Equation 6)). Taking into account the nearly Gaussian statistics of *x* and θ (at a given s≪l), we obtain $P\left(s\right)≃\frac{\sqrt{3}}{2\pi }$$\frac{1}{s}$, so
(68)$$\bar{n}\left(S\right)≃\frac{\sqrt{3}}{\pi }\ln\left(S/\lambda \right)$$

Equations (Equation 66) and (Equation 68) show that $p\left(s|0\right)∼\exp\left(-const\;\bar{n}\right)$ hinting at a Poisson-like distribution of *n*:(69)$$p\left(s|n\right)\equiv Z_{l}\left(s|n\right)/Z_{l}\left(s\right)≃{\left(s/\lambda \right)}^{-0.5}A^{n}/n!$$
with $A∼\bar{n}\left(s\right)$. Below, we derive the above heuristic Equation (Equation 69) in a rigorous way, clarifying its approximate nature and its region of validity $(n\ll A^{2/3}$). A more precise value of *A* is also obtained (cf. Equations (Equation 79) and (Equation 80)).

Let us turn to a more general case with penalized intersections (defined by the factor *w*) corresponding to the adsorption on a semi-permeable plane (i.e., a membrane with holes). In the general case (arbitrary *w*), the definition of $Z_{l}\left(s\right)$, Equation (Equation 64), must be amended to take into account that *n* crossings bring the statistical weight $w^{n}$:(70)$$Z_{l}\left(s,w\right)=\sum_{n}Z_{l}\left(s|n\right)w^{n}$$
where we explicitly show for clarity that $Z_{l}\left(s,w\right)$ depends also on the penetration parameter *w*. (Note that defining $Z_{l}\left(s|n\right)$ in this section, we do not apply any restriction on the angle $\theta_{cross}$ between the membrane plane and a polymer fragment intersecting it: the reason is that intersections only occur in the hole regions where the surface potential is absent).

The problem to find $Z_{l}\left(s,w\right)$ can be treated alternatively using the master Equation (Equation 35) (see Section 3). Using the same substitution as before $(\psi \left(x,\theta \right)=x^{\alpha }f\left(\eta \right)$, $\eta ={\left(l/x\right)}^{1/3}\theta$ for x>0; note that ψ for x<0 comes from the general relation (Equation 36) due to the obvious symmetry of the problem) we obtain $\frac{1}{\eta }f^{\prime \prime }+\frac{1}{3}\eta f^{\prime }-\alpha f=0$, which is equivalent to Equation (Equation 40).

Recalling that ψ(x,θ) is the partition function of a chain with the second end oriented at an angle θ to the plane and located at a distance $\left|x\right|$ from it, and comparing $\psi (0^{+},\theta )$ and $\psi (0^{-},\theta )$ for θ>0, we see that the only difference between $\psi (0^{+},\theta )$ and $\psi \left(0^{-},\theta \right)=\psi \left(0^{+},-\theta \right)$ is that $\psi (0^{+},\theta )$ involves one extra intersection near the end (cf. Figure 6). Hence
(71)$$\psi \left(0^{+},\theta \right)=w\psi \left(0^{-},\theta \right)=w\psi \left(0^{+},-\theta \right),\;\theta >0$$
where we also used Equation (Equation 36). (Note that physically $0^{+}$ corresponds to x=+Δ, $0^{-}$ to x=−Δ, and the condition θ>0 should be replaced with $x\gg \theta_{\Delta }$. Equation (Equation 71) is applicable as a boundary condition since we are interested in ψ(x,θ) at relatively large length-scales, $\left|x\right|\gg \Delta$). Obviously, for w=0, we return to the impenetrable wall problem considered in ref. [21], while w=1 corresponds to the case of a perfectly permeable surface (cf. Section 3). In the general case, the condition (Equation 71) can be rewritten in terms of the function f(η) as (cf. Equation (Equation 41)):(72)$$\lim_{\eta \to +\infty }\frac{f(\eta )}{f(-\eta )}=w$$

The solution of Equation (Equation 40) is
$$f\left(\eta \right)=\Psi \left(-\alpha ,2/3,-\eta^{3}/9\right)$$
where ψ(a,b,z) is the Tricomi function (cf. Equation (Equation 42)) defined in Section 3. Using Equation (Equation 43) and other general properties of Ψ, we obtain the following equation defining α based on Equation (Equation 72):sin(π(2/3+α))−sin(πα)=wsin(2π/3)

It has two physically acceptable solutions (corresponding to a ground-state f(η) without knots at finite η):(73)$$\alpha_{0}=\frac{1}{6}-\frac{1}{\pi }\arcsin\frac{w}{2},\;\alpha_{1}=-\frac{5}{6}+\frac{1}{\pi }\arcsin\frac{w}{2}$$

It is $\alpha =\alpha_{1}$ that corresponds to the critically adsorbed state (while $\alpha_{0}$ describes the non-adsorbed case corresponding to *u* = 0, i.e., a sterically repulsive membrane with no attraction).

The adsorbed layer can be considered as a system of loops. The number of loops of size *s* (per unit area) is proportional to $Z_{l}\left(s,w\right)$ so that the total length of monomers in all loops of size *s*, S/2<s<S is
$$L\left(S\right)\propto \int_{S/2}^{S}Z_{l}\left(s,w\right)sds∼S^{2}Z_{l}\left(S,w\right)$$

Most of these monomers are located at a distance $∼x\left(S\right)∼S^{3/2}/l^{1/2}$ leading to concentration
(74)$$c\left(x\right)∼L\left(S\right)/x\left(S\right)\propto S^{1/2}Z_{l}\left(S,w\right)\;\mathrm{at}x∼x\left(S\right)$$

Alternatively, the concentration profile can be obtained using the partition function ψ(x,θ) (cf. Equation (Equation 47)):$$c\left(x\right)=const\;\int \psi \left(x,\theta \right)\psi \left(x,-\theta \right)d\theta =const\;{\left|x\right|}^{2\alpha +1/3}\int f\left(\eta \right)f\left(-\eta \right)d\eta$$

The second integral in the above equation converges to a numerical factor if α<−1/6. The latter condition is always valid for $\alpha =\alpha_{1}$. Thus
(75)$$c\left(x\right)\propto {\left|x\right|}^{2\alpha_{1}+1/3}$$
at the adsorption threshold. Thus, once again we arrive at a self-similar concentration profile, $c\left(x\right)\propto {\left|x\right|}^{-\beta }$ in the proximal layer $\Delta ≲\left|x\right|≲l$ (cf. Equation (Equation 12) and ref. [21]), but now the critical exponent β depends on the membrane parameter *w*.

Comparing Equations (Equation 74) and (Equation 75) we finally obtain
(76)$$Z_{l}\left(s,w\right)∼{\left(s/\lambda \right)}^{3\alpha_{1}}$$

Obviously, for w=1, the above equation agrees with Equation (Equation 65), while for w=0, it agrees with the results of ref. [21] $(Z_{l}\propto s^{-5/2}$). Using Equations (Equation 76), (Equation 73) and (Equation 70) we obtain
(77)$$\sum_{n}Z_{l}\left(s|n\right)w^{n}=const\;{\left(s/\lambda \right)}^{-5/2}{\left(s/\lambda \right)}^{\left(3/\pi )\arcsin(w/2\right)}$$

Therefore, the statistical weights of loops with *n* intersections, $Z_{l}\left(s|n\right)$ can be obtained by just performing Taylor expansion in *w* of the rhs of the above equation. Moreover, this equation allows to easily obtain all the moments of the distribution $p\left(s|n\right)=Z_{l}\left(s|n\right)/Z_{l}\left(s,1\right)$. For example, the mean number of intersections is
$$\bar{n}=\partial \ln Z_{l}\left(s,w\right)/\partial \ln w=\frac{3}{\pi }\ln\left(s/\lambda \right)\frac{\partial }{\partial w}\left(\arcsin(w/2)\right)$$

For w=1, we thus obtain $\bar{n}=\frac{\sqrt{3}}{\pi }\ln\left(s/\lambda \right)$ in agreement with Equation (Equation 68). Using Equation (Equation 77), we find
(78)$$\sum_{n}p\left(s|n\right)w^{n}≃\;{\left(s/\lambda \right)}^{-0.5}e^{2A\arcsin(w/2)}$$
with
(79)$$A=\frac{3}{2\pi }\ln\left(s/\lambda \right)\gg 1$$

For a small w≲1/A the sum in the lhs of Equation (Equation 78) (which defines the generating function of the probability distribution) is dominated by a few terms with n∼1. Therefore, to obtain p(s|n) for not too large *n*, we can assume w∼1/A≪1 and approximate arcsin(w/2) as w/2 yielding
(80)$$\sum_{n}p\left(s|n\right)w^{n}≃\;{\left(s/\lambda \right)}^{-0.5}e^{Aw}$$

The above equation leads to the p(s|n) distribution announced in Equation (Equation 69). A simple analysis shows that higher-order terms in the expansion of the arcsin in Equation (Equation 78) can be indeed neglected if $n\ll A^{2/3}$. Note that the Poissonian character of the *n*-distribution is not self-evident since crossings of chain fragments with the plane are strongly correlated.

## 7. Discussion

### 7.1. Discussion on Two Models of Polymer/Membrane Interactions

In this paper, we consider the reversible adsorption of a wormlike polymer to an ultra-thin penetrable free-standing membrane focusing on the equilibrium structure of the adsorbed polymer layer. The adsorption is driven by a short-range attraction between the membrane surface and polymer segments. Two basic models of the membrane are considered: (i) a perfectly penetrable amphiphilic fluid layer (a surfactant bilayer) whose effect can be accounted for by a short-range membrane potential, Equation (Equation 4); and (ii) a partially permeable porous nanofilm (solid membrane with nano-holes), cf. Section 5 and Section 6. We found that in both cases, a strongly adsorbed anisotropic polymer layer (whose thickness is smaller than the Kuhn segment length *l*) is formed near the membrane slightly above the critical attraction strength $u^{*}$. In this regime, the number $n_{c}$ of polymer/membrane contacts is roughly proportional to the polymerization degree (see Equation (Equation 15) and the text below it). The parameter $u^{*}$ always scales with the attraction range Δ according to Equation (Equation 5). It shows that $u^{*}$ decreases both with Δ and with the chain stiffness defined by the Kuhn segment *l*. Equation (Equation 5) agrees with numerous theoretical and simulation results for polymer adsorption to an impenetrable surface [18,21,27,46,47]. Thus, although $u^{*}$ must be lower for a penetrable film, the scaling relation, Equation (Equation 5), seems to be universal.

For both models, the adsorbed layer is symmetric: it is formed on both sides of the film. Generally, an adsorbed layer involves trains, loops and tails (Figure 1b). Trains are chain fragments (of length $s_{train}≳\lambda$) located mostly within the attraction layer (of thickness ∼Δ). Loops are fragments between two trains located mostly outside the Δ-layer. Since the loop ends are connected to trains, they must be oriented almost tangentially to the attracting surface (with a small tilt angle θ, $\theta ≲\theta_{\Delta }∼{\left(\Delta /l\right)}^{1/3}$, cf. Equation (Equation 37)). A tail is a terminal chain part with a free end (which is typically located outside the Δ-layer) whose starting point is adjacent to a train. Tails are not important for long chains above the critical point $(u>u^{*}$), where the adsorbed layer consists mostly of trains and loops. There are two types of loops: large semiflexible isotropic loops (of length s≫l) and shorter aligned loops (s≲l) forming two sublayers, distal ($\left|x\right|\gg l$) and proximal ($\Delta ≲\left|x\right|≲l$), respectively. At $u\approx u^{*}$, aligned loops in the proximal layer show a fractal size distribution leading to an algebraic decay of polymer concentration $c\left(x\right)\propto {\left|x\right|}^{-\beta }$. For model ‘i’, the number density of aligned loops of length *s* is $\rho \left(s\right)\propto Z_{l}\left(s\right)\propto s^{-2}$ (Equation (Equation 11)) and β=1 (Equation (Equation 12)). For model ‘ii’, the critical exponents depend on the area fraction *w* of the holes. We found that in the proximal layer, $\rho \left(s\right)\propto Z_{l}\left(s,w\right)\propto s^{3\alpha_{1}}$ (Equation (Equation 76)), where $\alpha_{1}=-\frac{5}{6}+\frac{1}{\pi }\arcsin\frac{w}{2}$ (Equation (Equation 73)), and that
(81)$$\beta =-\frac{1}{3}-2\alpha_{1}$$ (Equation (Equation 75)). Thus, for model ‘ii’ with 0≤w<1 we get 4/3>β>1. For w=0, the critical exponents coincide with the results for impenetrable surface [21]: β=4/3, $3\alpha_{1}=-5/2$. The opposite limit, w→1, corresponds to a perfectly permeable film. It is, therefore, not surprising that model ‘ii’ reproduces the exponents of model ‘i’ in the w→1 regime.

There is, however, a qualitative difference between the two models: Model ‘i’ is characterized by two adsorption transitions: at u=0 (weak isotropic adsorption) and at $u=u^{*}$ (strong anisotropic adsorption). The first transition results in formation of a thick layer of long semiflexible isotropic loops. Its thickness at $0<u\ll u^{*}$ is (cf. Section 4)
$$h∼l{\left(l/\Delta \right)}^{1/3}u^{*}/u\gg l$$

However, this layer rapidly thins as *u* increases tending to $u^{*}$ (see the first two regimes in Equation (Equation 61)). At the same time, the self-similar proximal layer of aligned loops rapidly grows so that near $u^{*}$ the proximal layer (with concentration profile $c\left(x\right)\propto 1/\left|x\right|$ for $\left|x\right|≲l$) becomes well-developed and dominant, marking the second transition occurring in a narrow region $\left|\tau \right|=\left|u/u^{*}-1\right|≲{\left(\Delta /l\right)}^{2/3}$. Adsorption at a symmetric interface (for a model analogous to our model ‘i’) was considered theoretically in ref. [48]. In the weak adsorption regime (apparently corresponding to *u* well below $u^{*}$), two length-scales characterizing the concentration profile (one ∼l and another much longer) were identified [48], in qualitative agreement with our results.

By contrast, model ‘ii’ shows only one adsorption transition at $u=u^{*}$, leading to the formation of a self-similar proximal layer of aligned loops at $0<\tau ≲{\left(\Delta /l\right)}^{2/3}\ll 1$. As τ is further increased, the proximal layer thins in a similar way for both models (as was established following the approach of Section 4); the thinning law for model ‘ii’ reads
(82)$$h∼\Delta /\tau^{3/2},\;1≳\tau ≳{\left(\Delta /l\right)}^{2/3}$$ (cf. Equation (Equation 63)). The adsorbed layer thickness becomes comparable to the membrane attraction range, h∼Δ, at τ∼1. Note that Equation (Equation 82) just slightly differs from the prediction, Equation (Equation 63), for model ‘i’: an additional log-factor, ${\left[\ln(1/\tau )\right]}^{3/2}$, is present in the latter case.

### 7.2. Notes Related to the Model of Solid Film with Holes

In Section 6, we neglected both the chain thickness *d* and the thickness $d_{m}$ of solid parts of the membrane (shown in black in Figure 5). With finite *d* and $d_{m}$ the chain penetration through a hole becomes slightly restricted: For geometric reasons, a penetration is possible if the tilt angle $\left|\theta \right|$ of the relevant chain segment exceeds $\theta_{min}=\left(d+d_{m}\right)/D$. Obviously, this condition is virtually always satisfied if the typical tilt angle $\left|\theta \right|∼{\left(s/l\right)}^{1/2}$ (for a loop of length *s*) is much larger than $\theta_{min}$, that is, if
(83)$$s/l\gg {\left(d+d_{m}\right)}^{2}/D^{2}$$

The latter condition defines the lower bound for the range of loop sizes where the theory of Section 6 (including the scaling relations) is applicable. Recalling that s≪l, the condition (Equation 83) demands that $D\gg d+d_{m}$, which is always satisfied if $d+d_{m}≲\Delta$ and D≫Δ as assumed at the beginning of Section 6. Note also that for D≳λ (which is equivalent to the condition $D/\sqrt{w}≳\lambda$ assumed in Section 6, provided that *w* is not small) and $d+d_{m}≲\Delta$ the basic condition (Equation 83) simply follows from the condition s≫λ, which is always required for the loops to follow a self-similar distribution in the proximal layer.

Let us turn to the size (length *s*) distribution of trains. For an impermeable membrane (w→0), the probability density of the train length *s* is nearly exponential for s≳λ. It follows from the fact that train sections of size Δs∼λ are essentially decorrelated, and hence the probability $p_{l}$ that the next segment Δs escapes the attraction zone (the Δ-layer), and thus creating a (short) loop is independent of the train length. Note that by the definition of the length-scale λ, the probability $p_{l}∼1/2$, and hence the mean train length is $s_{train}∼2\lambda$.

For w>0, the holes do not perturb the train distribution provided that the typical size of solid parts of the membrane $D_{s}=D/\sqrt{w}-D$ is much larger than $s_{train}$, $D_{s}\gg s_{train}$, which is equivalent to
(84)D(1−w)≫λ

Thus, the train length distribution remains exponential also for a porous membrane with sufficiently large holes (obviously, this condition becomes stronger as *w* increases, w→1). Note, however, that the condition (Equation 84) is irrelevant to the self-similar distribution of loops considered in Section 6, which remains intact as long as D(1−w)≳λ. It is the critical adsorption strength $u^{*}$ that becomes affected by *D* and *w* in the regime D(1−w)≲λ: $u^{*}$ increases as *D* is decreased or *w* is increased because the train size must be reduced here for geometric reasons.

### 7.3. Loop and Tail Distributions

Let us turn to the effect of the chain contour length *L*. Incidentally, we assumed that L≲l in Section 2.2 and Section 2.3 where the monomer distribution in the adsorbed layers is considered using scaling arguments, while long chains, L≫l, are assumed in Section 3, Section 4 and Section 6, presenting the quantitative theory. It is noteworthy that short chains (L≲l) cannot be trapped by a membrane at $u<u^{*}$: they would escape into the solution bulk. That is why we had to assume (in Section 2.2 and Section 2.3) that the chain ends are forced to be located in the film. The only significant implication of this condition is that the terminal layer thickness for a moderate *u* is defined by the chain length, $x_{max}∼L^{3/2}/l^{1/2}$ (cf. Equation (Equation 13)), rather than by the attraction strength *u*, as it happens in the case of very long chains, L→∞ (cf. Equations (Equation 61) and (Equation 63)). As a result, the distal layer is totally absent for short chains, L≲l. However, the self-similar structure of loops in the proximal layer is virtually independent of *L* above $u^{*}$: it is nearly the same for L≫l and L∼l. In the general case, a finite chain length *L* means a finite contribution of tails to the polymer concentration profile c(x). Noteworthily, the tail effect was avoided so far by either ‘pinning’ both chain ends or considering long chains, L→∞. Below, we analyze the tail effect for short chain L≲l pinned at only one end, thus allowing for one tail. (Instead of pinning, one can also demand a sufficiently high adsorption strength to trap the chain near the membrane; both methods give similar results).

Let $Z_{1}\left(L\right)$ be the total partition function of a chain (of length λ≪L≲l) with one end free and another end pinned to the membrane (with tangential orientation of the end segment), and $Z_{2}\left(L\right)$ be a similar partition function with the condition that both ends are located in the membrane attraction layer (allowing one end to freely glide along the surface to anneal the end-to-end distance fluctuations). Let us focus on the model ‘ii’ (solid film with holes) at the critical point $u=u^{*}$. The concentration due to loops at this point follows the power law, $c\left(x\right)\propto x^{-\beta }$ with β>1. Therefore ∫c(x)dx is dominated by short x∼Δ, so that a finite fraction of polymer segments is inside the Δ-layers near the membrane. In this case, $Z_{2}\left(2s\right)≃const\;Z_{2}{\left(s\right)}^{2}$ since the probability to find a train (a segment of length ≳λ in contact with the membrane) near the middle of the 2s-chain is close to 100%. The above relation leads to $Z_{2}\left(s\right)≃const\;$$e^{\epsilon s}$, where ϵ is the adsorption energy per unit length, which must be equal to 0 by definition of the adsorption threshold $u^{*}$. Therefore, $Z_{2}\left(s\right)≃const\;$. Turning to $Z_{1}\left(L\right)$ we note that any configuration of wormlike chain (with one end pinned and another free end) can be uniquely divided in two fragments: the first one, of length L−s with ends in tangential contact with the membrane, and the second, a tail of length *s* (with no trains in it). This decomposition leads to the following equation
(85)$$Z_{1}\left(L\right)≃const\;\int_{0}^{L}Z_{2}\left(L-s\right)Z_{t}\left(s\right)ds$$
where $Z_{t}\left(s\right)$ is the tail partition function. Since a loop of length 2s can be considered as a combination of two tails, there is a general relation between $Z_{t}\left(s\right)$ and $Z_{l}\left(2s\right)$ (cf. ref. [21]):(86)$$Z_{l}\left(2s\right)∼Z_{t}{\left(s\right)}^{2}/\left(x\left(s\right)\theta \left(s\right)\right)$$
where $x\left(s\right)\propto s^{3/2}$ (cf. Equation (Equation 6)) and $\theta \left(s\right)\propto s^{1/2}$ for s≪l. Recalling that $Z_{l}\left(s\right)\propto s^{3\alpha_{1}}$ (cf. Equation (Equation 76) and note that $Z_{l}\left(s\right)=Z_{l}\left(s,w\right)$ in this section, i.e., the parameter *w* is omitted for brevity) we obtain
(87)$$Z_{t}\left(s\right)\propto s^{1+3\alpha_{1}/2},\;s≲l$$
where (cf. Equation (Equation 81))
(88)$$\gamma =1+3\alpha_{1}/2=\left(3/4\right)\left(1-\beta \right)$$

Equation (Equation 85) implies that at $u=u^{*}$ the statistical weight of all conformations with tail of length *s* is proportional to $Z_{2}\left(L-s\right)Z_{t}\left(s\right)$, and hence the tail length distribution density is
(89)$$\rho_{t}\left(s\right)≃Z_{2}\left(L-s\right)Z_{t}\left(s\right)/\int_{0}^{L}Z_{2}\left(L-s\right)Z_{t}\left(s\right)ds$$

Finally, using Equation (Equation 87), we obtain (for L≪l)
$$\rho_{t}\left(s\right)≃s^{\gamma }\left(\gamma +1\right)/L^{\gamma +1}$$

The mean tails length $L_{t}=\langle s\rangle$ is then given by
$$L_{t}/L≃\frac{\gamma +1}{\gamma +2}=\frac{4+3\alpha_{1}}{6+3\alpha_{1}},\;L\ll l$$

Recalling that $-2/3>\alpha_{1}\geq -5/6$ (cf. Equation (Equation 73)), we conclude that in critical conditions $(u=u^{*}$) the mass fraction of tails ($L_{t}/L$) is around 50%, and therefore it is always comparable to the mass fraction $(=\left(L-L_{t}\right)/L$) of loops and trains (in addition, contributions of loops and trains are obviously comparable as well). A similar argument also shows that mass fractions of trains in the case of one pinned end $(w_{train1}$) and for two ends trapped near the membrane $(w_{train2}$) are related
(90)$$w_{train1}≃\frac{1}{\gamma +2}w_{train2}$$

Thus, in effect, the above results based on Equation (Equation 85) show that the tail length is typically ∼L/2 at $u=u^{*}$. The monomer distribution due to such a tail is therefore comparable with that for a single tail of length *L*. The latter distribution follows from Equation (Equation 6) (for x>0):(91)$$c_{t}\left(x\right)∼{\left(l/x\right)}^{1/3},\;x≲x_{max},\;x_{max}∼L^{3/2}/l^{1/2},\;L≲l$$ (Note that here and below we define $c_{t}\left(x\right)$ in such a way that $c_{t}\left(x\right)dx$ is the mean length of tail segments in the layer (x,x+dx); a similar definition applies to $c_{l}\left(x\right)$ for loops and trains). At $u=u^{*}$, the concentration contribution of loops/trains is
(92)$$c_{l}\left(x\right)∼\frac{L}{\Delta }{\left(\frac{\Delta }{x}\right)}^{\beta },\;\Delta ≲x≲x_{max}$$
where β is defined in Equation (Equation 81) (cf. also Equation (Equation 75)). The prefactor in Equation (Equation 92) can be obtained from the condition of nearly equal masses of loops/trains and tails, $\int c_{l}\left(x\right)dx∼\int c_{t}\left(x\right)dx∼L/2$, as discussed above. Note also that both $c_{t}$ and $c_{l}$ vanish exponentially at $x>x_{max}$.

Obviously, loops dominate at short distances, while tails are more important at the periphery:(93)$$c\left(x\right)∼\left\{\begin{array}{cc} c_{l}\left(x\right),\; & \Delta <x<x^{*}\\ c_{t}\left(x\right), & x_{max}≳x>x^{*} \end{array}\right.$$
where $c\left(x\right)=c_{l}\left(x\right)+c_{t}\left(x\right)$ is the total concentration, and
(94)$$x^{*}∼\Delta {\left(L/\lambda \right)}^{1/(\beta -1/3)}∼\Delta {\left(x_{max}/\Delta \right)}^{2/(3\beta -1)}$$
is the crossover distance; $\Delta \ll x^{*}<x_{max}$ since 3β−1>2 and L≫λ. (Note that we do not consider trains any more focusing on the regime x>Δ; it must be always kept in mind that an equivalent adsorbed layer at x<−Δ is also present). Equation (Equation 94) is applicable for short chains, L≲l. For L∼l, it gives
(95)$$x^{*}∼l{\left(\Delta /l\right)}^{\frac{\beta -1}{\beta -1/3}}≳l^{2/3}\Delta^{1/3}$$

Qualitatively the same picture stays valid also for longer chains, L≫l. In this case, the chain behaves flexibly at length scales x≫l, so $x_{max}∼{\left(Ll\right)}^{1/2}$ (i.e., $x_{max}$ is defined by the coil size *R*, Equation (Equation 3)). At the critical adsorption point $(u=u^{*}$), the concentration contribution of semiflexible isotropic loops is nearly constant in the regime x≫l (following the classical results on the ideal flexible chain adsorption [4]):(96)$$c_{l}\left(x\right)∼\left\{\begin{array}{cc} c_{0}, & x_{max}∼{\left(Ll\right)}^{1/2}>x>l\\ c_{0}{\left(l/x\right)}^{\beta }, & \Delta <x<l \end{array}\right.$$
where $c_{0}∼L/\left(\sqrt{Ll}+l^{\beta }\Delta^{1-\beta }\right)$ is found from the basic principle that total masses of loops and tails are always comparable at $u=u^{*}$.

As for the tails, their monomer concentration profile is defined in Equation (Equation 91) for x≲l, while $c_{t}\left(x\right)\propto x$ for x≳l (note that the membrane acts as a *repulsive* surface for long semiflexible tails since by definition the tail monomers never enter the surface attraction zone):(97)$$c_{t}\left(x\right)∼\left\{\begin{array}{cc} x/l, & x_{max}>x>l\\ {\left(l/x\right)}^{1/3}, & \Delta <x<l \end{array}\right.$$

Note that the scaling law, $c_{t}\left(x\right)\propto x$ at x≫l (valid for a tail emanating from a repulsive surface) comes from a simple argument already used to derive Equation (Equation 91): $c_{t}\left(x\right)∼\frac{s/2}{x/2}$, where s/2 is the length of tail fragment between points s/2 and *s* along its contour, and x/2 is the thickness of the layer where this fragment is typically located spanning distances between x/2 and *x*, with $x∼\sqrt{sl}$ being its typical size, cf. Equation (Equation 3).

Using Equations (Equation 96) and (Equation 97), we find that monomers in the loops dominate at $x<x^{*}$, and tail monomers - at $x>x^{*}$ (cf. Equation (Equation 93)), where
(98)$$x^{*}∼\left\{\begin{array}{ccc} l{\left[L/\left(l^{\beta }\Delta^{1-\beta }\right)\right]}^{1/(\beta -1/3)} & , & L<l^{\beta }\Delta^{1-\beta }\\ lL/\left(\sqrt{Ll}+l^{\beta }\Delta^{1-\beta }\right) & , & L>l^{\beta }\Delta^{1-\beta } \end{array}\right.$$

Thus, the tail contribution in the proximal layer can be totally neglected if $L\gg L^{*}∼l^{\beta }\Delta^{1-\beta }$. Note that $L^{*}\gg l$ since β>1. The obtained results can be applied to the case of an impenetrable surface as well by setting β=4/3$(\alpha_{1}=-5/6$). The crossover distance $x^{*}$ always increases with *L* (cf. Equations (Equation 94) and (Equation 98)).

The energy *E* of the polymer/membrane complex is proportional to the number of polymer/surface contacts $\left(n_{c}\right)$ or, in other words, to the total polymer length $L_{c}$ in the attractive layers of thickness Δ,
$$L_{c}=2\int_{0}^{\Delta }c\left(x\right)dx∼c\left(\Delta \right)\Delta$$

Taking into account that $c\left(\Delta \right)∼c_{l}\left(\Delta \right)$ and using Equations (Equation 92) and (Equation 96), we obtain $L_{c}$ at the critical point $u=u^{*}$:(99)$$L_{c}∼\left\{\begin{array}{ccc} L/2 & , & L<\tilde{L}\\ \sqrt{L\tilde{L}} & , & L>\tilde{L} \end{array}\right.$$
where the crossover length $\tilde{L}$ is
(100)$$\tilde{L}∼l{\left(l/\Delta \right)}^{2(\beta -1)}\gg L^{*}\gg l$$

The fraction $p=L_{c}/L$ of trapped monomers, therefore, is (cf. Equation (Equation 59)):(101)$$p∼\left\{\begin{array}{ccc} 0.5 & , & L<\tilde{L}\\ \sqrt{\tilde{L}/L} & , & L>\tilde{L} \end{array}\right.$$

Equation (Equation 99) shows that the contact exponent ϕ (cf. Equation (Equation 1)) changes from ϕ=1 for relatively short chains $(L\ll \tilde{L}$) to ϕ=1/2 for longer chains $\left(L\gg \tilde{L}\right)$. It is remarkable that ϕ=1 is thus predicted also for semiflexible chains $\left(\tilde{L}\gg L\gg l\right)$ forming a complicated structure of competing long loops, aligned loops and tails. The result, ϕ=1/2, for very long chains (L→∞) trivially corresponds to the well-known critical behavior of ideal flexible polymers [4]. However, even in this long-chain limit, the amount of contacts rapidly increases from $L_{c}\propto L^{1/2}$ to the maximum $L_{c}∼L$ as the membrane attraction strength increases above $u^{*}$. Indeed, at $u>u^{*}$, the adsorption free energy ϵ is positive (cf. Equation (Equation 34)), so the loops and tails of length s≫1/ϵ are exponentially suppressed (cf. Equation (Equation 26)). Therefore, the concentration profile for L→∞ at a given $\tau =u/u^{*}-1>0$ becomes roughly equivalent to the profile of loops at the critical point for L∼1/ϵ, where ϵ=ϵ(τ). Using Equation (Equation 101) we thus obtain:(102)$$p∼\left\{\begin{array}{ccc} 0.5 & , & \epsilon \tilde{L}>1\\ \sqrt{\epsilon \tilde{L}} & , & \epsilon \tilde{L}<1 \end{array}\right.$$

It remains to find out how ϵ depends on the deviation $\tau =u/u^{*}-1>0$ from the critical point. Using Equations (Equation 25) and (Equation 102) we obtain the following as a result:(103)$$\epsilon ∼\left\{\begin{array}{ccc} \tau^{2}\tilde{L}{\left(u^{*}\right)}^{2} & , & \tau ≲\tau^{*}\\ \tau u^{*} & , & \tau ≳\tau^{*} \end{array}\right.$$
(104)$$p∼\left\{\begin{array}{ccc} \tau /\tau^{*} & , & \tau ≲\tau^{*}\\ 0.5 & , & \tau ≳\tau^{*} \end{array}\right.$$
where $\tau^{*}∼1/\left(u^{*}\tilde{L}\right)∼\lambda /\tilde{L}∼{\left(\Delta /l\right)}^{2\beta -4/3}$. Note that $\tau^{*}\ll 1$ since 2/3<2β−4/3≤4/3. Equation (Equation 104) therefore shows that a small increase of *u* from $u^{*}$ to $u^{*}\left(1+\tau^{*}\right)$ leads to a strong increase of the contact fraction *p* from 0 to p∼1 implying that $L_{c}∼L$ as stated above. The terminal thickness *h* of the adsorbed layer is nearly equal to the extension (in the direction perpendicular to the membrane) of the terminal loops of contour length s∼1/ϵ. Thus, using Equation (Equation 103) we obtain
(105)$$h∼\left\{\begin{array}{ccc} l{\left(\Delta /l\right)}^{\beta -1/3}\tau^{-1} & , & 0<\tau <\tau^{*}\\ l{\left(\Delta /l\right)}^{1/3}\tau^{-1/2} & , & \tau^{*}<\tau <\tau^{**}\\ \Delta /\tau^{3/2} & , & \tau^{**}<\tau <1 \end{array}\right.$$
where $\tau^{**}∼\delta^{2}={\left(\Delta /l\right)}^{2/3}$. In particular, we get h∼l for $\tau ∼\tau^{**}$. Therefore, the distal layer is suppressed at $\tau >\tau^{**}$.

### 7.4. Non-Ideality Effects

In this study, we assumed no interactions between polymer segments. This assumption is justified for a sufficiently small ratio of polymer chain thickness *d* to its Kuhn segment *l*, d/l≪1, which ensures that the volume fraction occupied by a single chain is low. The bottleneck for this assumption occurs in the contact Δ-layer near the membrane, where the polymer concentration (volume fraction $c_{\phi }$) increases in the adsorbed state $(u\geq u^{*}$). The maximum polymer volume fraction $c_{\phi }$ there can be estimated as $c_{\phi }∼Ld^{2}/\left(\Delta R^{2}\right)∼d^{2}/\left(l\Delta \right)$, where *R* is the coil size defined in Equation (Equation 3). The excluded-volume interaction free energy per Kuhn segment is $U_{l}∼Tc_{\phi }l/d∼Td/\Delta$ [49]. Therefore, for Δ≫d, the interactions are rather weak, $U_{l}\ll T$, and can be neglected for the adsorption profile c(x) and the contact fraction *p* (at least, in the most interesting regime $\tau \gg \delta^{2}d/\Delta$). By contrast, interactions of segments are important if the membrane potential is very short-range, Δ≲d. In this case, the adsorbed chain size in the yz-plane must be significantly swollen compared to the ideal size, Equation (Equation 3). This effect was observed in simulation studies [27,50,51] and experimentally for DNA molecules electrostatically bound to cationic lipid bilayers [10]. (Note that the two-dimensional swelling effect leads to a decrease in $c_{\phi }$ and, therefore, to a lower free energy contribution due to interactions). In both cases, however, the interactions between monomers either in the Δ-layer or outside it are just marginally important for the statistics of long loops (provided that the Fixman parameter $z_{F}∼\left(d/l\right){\left(L/l\right)}^{1/2}$ [52] is sufficiently low, that is $L\ll l^{3}/d^{2}$). The main effect of the interactions for the adsorption transition then reduces to some renormalization (an increase) of the critical adsorption strength $u^{*}$.

### 7.5. Estimates of the Layer Thickness

Here, we provide some estimates on the distance range corresponding to the self-similar structure of aligned loops (which is the most interesting part of the adsorbed layer at the critical point) for some hypothetical polymer/membrane systems. As examples, let us consider two types of membranes: a typical phospholipid membrane of effective thickness 2Δ∼8 nm in water and a porous carbon nano-membrane of thickness $d_{m}∼1$ nm with holes of diameter D∼10 nm [45]. As polymers, we can consider a double-strand DNA (with thickness d∼2 nm and Kuhn segment $l=2l_{p}∼100$ nm) and chitosan with d∼1 nm and $l=2l_{p}∼90$ nm (at a low salt concentration $c_{s}=1.25$ mM [53]; $l_{p}$ for chitosan decreases at higher $c_{s}$) in aqueous solutions. The case of lipid membrane roughly corresponds to the model of Section 2.2 and Section 3 (here, we set aside issues related to whether polymer penetration through the membrane is kinetically allowed or not), so the self-similar sublayer is expected in the distance range $\Delta <\left|x\right|<l$ which corresponds to $4\;nm<\left|x\right|<100\;nm$ for both polymers. For the porous carbon nano-membrane in water, the attraction range is defined mostly by the Debye length $r_{D}$, $\Delta ∼r_{D}$. This Δ should be compared with $x_{min}∼D^{3/2}/l^{1/2}$ for a moderate porosity w∼0.5 (cf. Section 6). For both polymers (DNA and chitosan), we thus arrive at $x_{min}∼3$ nm. In the case of DNA adsorption, we assume physiological ionic strength, $c_{s}\approx 0.15$ M, so $\Delta ∼r_{D}∼1$ nm. Therefore, for the DNA adsorption onto the carbon membrane $x_{min}>r_{D}$, and hence a self-similar structure of loops is predicted for $x_{min}<\left|x\right|<l$, i.e., for $3\;nm<\left|x\right|<100\;nm$. By contrast, for chitosan adsorption (at $c_{s}=1.25$ mM) the screening length is much longer, $r_{D}∼9\;nm$, hence $r_{D}>x_{min}$ and the sublayer of aligned loops corresponds to $r_{D}<\left|x\right|<l$, i.e., to $\left|x\right|$ between 10 and 100 nm.

## 8. Summary

We theoretically analyzed adsorption of a stiff wormlike polymer chain (with large persistence length) on permeable and semi-permeable free-standing flat nano-membranes. The membrane attraction length-range Δ is assumed to be much shorter than the polymer Kuhn segment *l*. Two membrane models are considered: In the first case, adsorption is driven by monomer attraction to a uniform membrane (modeled as a short-range potential field). A nanoporous solid film whose surface attracts polymer segments is studied as the second model (its relevant example is described in ref. [45]). In both cases, we predict the formation of a fractal proximal layer of aligned polymer loops at the critical strength $u^{*}$ of monomer/film attraction ($u^{*}$ is approximately given by Equation (Equation 5) for both models). The monomer concentration *c* strongly increases near the membrane according to a power law $c\left(x\right)\propto x^{-\beta }$ in the proximal layer. It is shown that the critical exponent β for the first model (β=1) is different from the value obtained for adsorption to an impenetrable surface (β=4/3). For the second model, we demonstrate that β depends on the area fraction of holes, *w*; the universal asymptotically exact dependence β=β(w) is obtained analytically (cf. Equation (Equation 81)). Moreover, we characterize both spatial and orientational distributions of polymer segments in the proximal layer in an asymptotically exact way.

We established that generally, the adsorbed layer is characterized by a complex structure involving (on each side of the membrane) the thin contact Δ-layer of trapped segments, the fractal proximal layer of thickness comparable with *l* (or smaller at $u>u^{*}$), and the distal layer of long semiflexible loops. Moreover, polymer loops and tails can compete in both proximal and distal layers. We found that for the first model, the distal layer is suppressed already at $u=u^{*}$, while for the second model, this happens slightly above the threshold $u^{*}$, at $u/u^{*}-1∼{\left(\Delta /l\right)}^{2/3}$ (cf. Equation (Equation 105)). At a higher attraction strength *u* for both models, the adsorbed layers can be viewed as a fractal distribution of loops. It is also shown that the fraction *p* of trapped chain segments in contact with the membrane becomes significant, p∼0.5, for $u/u^{*}-1≳{\left(\Delta /l\right)}^{2\beta -4/3}$, again for both models.

Turning to the contribution of polymer tails, we established that it is not important for very long chains. However, in the general case of finite chain length *L*, the tails can dominate the adsorbed polymer concentration profile at large distances, $x>x^{*}$, while loops dominate closer to the membrane. Near the adsorption threshold, $u\approx u^{*}$, the crossover distance $x^{*}$ increases with *L* as $x^{*}\propto L^{1/(\beta -1/3)}$ for $L<L^{*}$, $x^{*}\propto L$ for $L^{*}<L<{\left(L^{*}\right)}^{2}/l$, and $x^{*}\propto \sqrt{L}$ for $L>{\left(L^{*}\right)}^{2}/l$, where $L^{*}∼l{\left(l/\Delta \right)}^{\beta -1}$ (cf. Equation (Equation 98)). As a side product, we obtained the generating function of the probability distribution of the number of intersections between a geometric plane and a wormlike chain, whose ends are tangentially attached to the plane (cf. Equation (Equation 78)).

We thus provide a powerful theoretical method (including both the quantitative transfer-matrix approach and scaling concepts) to study the microstructure of adsorbed wormlike polymer layers. The current work is therefore likely to open new perspectives in predicting the physical properties of various polymer/membrane complexes. The significance of the obtained results is underlined by the facts that semiflexible polymers are often used in applications to modify the properties of membranes and films and to tune the interfacial properties of functional layers [1,2,3,4,5,6,7,33,34], that the adsorption of semiflexible macromolecules on substrates is involved in many biological processes [8,9], that important biopolymers, such as double-strand DNA, polypeptides and protein biofilaments, can be roughly described by the wormlike chain model [10,25,26,54,55], and that their interactions with cell membranes are of key interest in understanding various biological functions of living systems [8,9,10,11,34,35,36,37,38].

## Figures and Tables

**Figure 1 polymers-15-00035-f001:**
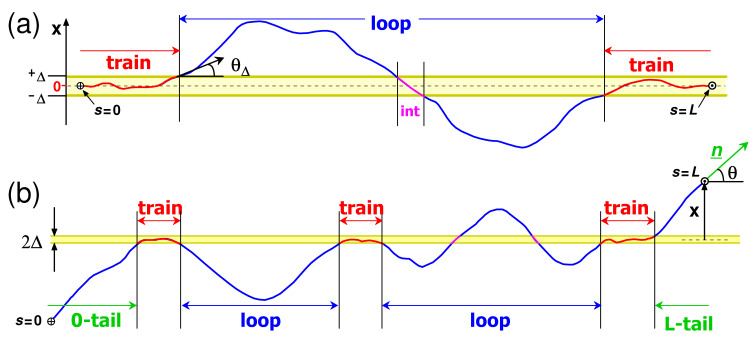
(**a**) A wormlike chain with two ends located in the membrane (at −Δ<x<Δ). The chain segments are oriented at small angles θ to the membrane plane (x=0). Typically, $\left|\theta \right|≲\theta_{\Delta }$ in the ‘train’ sections, and $\left|\theta \right|\gg \theta_{\Delta }$ in the ‘loops’ (see Equation (Equation 10)). (**b**) A wormlike chain of contour length *L* with free ends; the chain is aligned by attraction to a penetrable membrane. The right-hand end is located at distance *x* from the membrane mid-plane; its orientation is defined by the unit vector $\underline{n}$. The chain involves ‘train’ sections trapped in the short-range attraction layer representing the membrane, ‘loops’ of different sizes away from it, and two ‘tails’.

**Figure 2 polymers-15-00035-f002:**
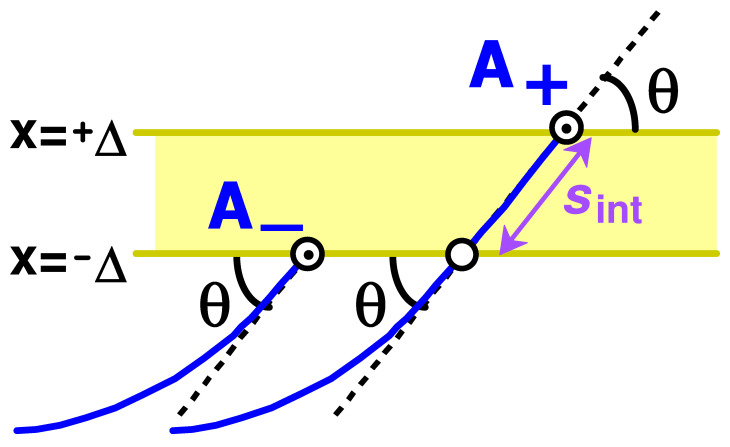
Two chains with the same orientation at x=−Δ defined by the angle θ>0. One chain ends at point $A_{-}$ just below the membrane. The second chain ends at point $A_{+}$ slightly above the membrane intersecting it with the end segment of length $s_{int}$.

**Figure 3 polymers-15-00035-f003:**
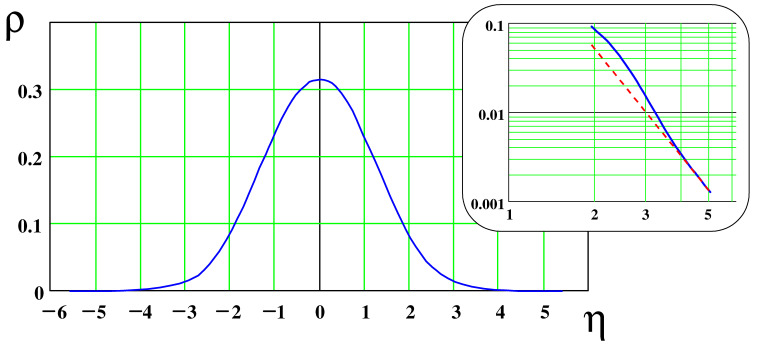
The normalized distribution (see Equation (Equation 49)) of the reduced angle $\eta =\theta {\left(l/x\right)}^{1/3}$ in the proximal aligned layer, where θ is the angle of a chain segment and *x* is its distance to the membrane. Inset: The same function in the log-log scale. The dashed line indicates the asymptotic law, $\rho \propto \eta^{-4}$ for $\left|\eta \right|\gg 1$ (cf. Equation (Equation 51)).

**Figure 4 polymers-15-00035-f004:**
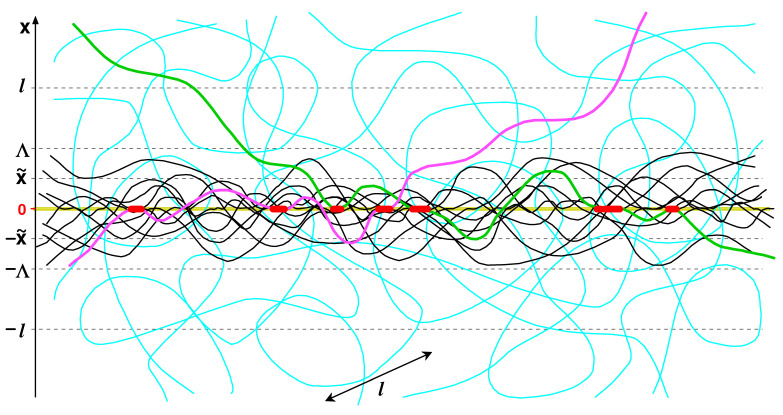
The structure of loops formed by a wormlike chain near a penetrable membrane at a near-critical adsorption strength, $u\approx u^{*}$, $u<u^{*}$. There are long semiflexible isotropic loops (shown in cyan) everywhere, while shorter aligned loops (shown in black) are present only close enough to the membrane. Aligned loops show a fractal distribution for $\left|x\right|≲\Lambda$, and they dominate by mass in the proximal layer at $\left|x\right|≲\tilde{x}$, where $\tilde{x}<\Lambda$. The aligned loops virtually merge with long isotropic loops at the crossover distance $\left|x\right|∼\Lambda$ (see green and pink fragments as examples).

**Figure 5 polymers-15-00035-f005:**
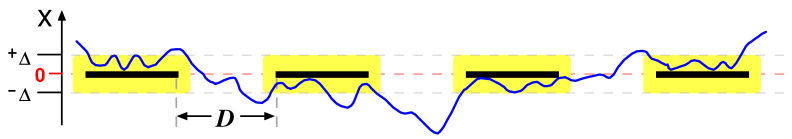
A wormlike chain near a porous solid nanofilm of effective thickness 2Δ. The chain can freely pass from one side of the film to the other through the holes of diameter ∼D.

**Figure 6 polymers-15-00035-f006:**
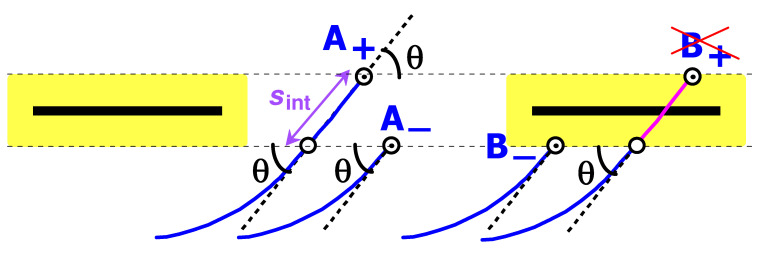
Clarification on Equation (Equation 71): The chain ending at point $A_{+}$ above the membrane intersects it through a hole. Otherwise, the conformation of this chain is similar to that of the chain $A_{-}$ which ends below the membrane thus avoiding the last intersection. In analogy with Figure 2 the partition functions (ψ) of the chains $A_{-}$ and $A_{+}$ are nearly equal, and the same is true, of course, for the chains $A_{-}$ and $B_{-}$. By contrast, the partition function of the chain $B_{+}$ is zero since intersections of a solid part of the membrane are not allowed. On averaging over the film area, these relations lead to Equation (Equation 71).

## Data Availability

Not applicable.

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
