# Peer review of "Adsorption of Wormlike Chains onto Partially Permeable Membranes"

_polymers, 2022, doi:10.3390/polym15010035_

Round 1
Reviewer 1 Report
The paper is the theoretical investigation of the polymers molecules behavior in contact with the (semi)permeable membrane. The authors develop the previously published model for the polymer sorption on the solid surface. The received results may be utilized in the modelling of the membranes properties in different applications, such as membrane modification, the changes of the membrane properties during the petroleum filtration and so forth. There are following comments to the paper:
1) The paper will be significantly improved with demonstration of the comparison of the results of the suggested model to the practical values (may be via estimation of the thickness of the adsorbed layer of the polymer), as the authors suppose that suggested mathematical apparatus might be used for modelling the great variety of polymer-membrane systems.
2) The explication section should be added with all the parameters used in the models.
3) The authors should more clearly distinct the previous results concerning the sorption of the solid surface and the current model for polymer-permeable membrane systems to avoid the repetition of the results of paper [21] in some sections.
Author Response
We thank the reviewer for useful comments. Concerning their points:
Point 1: We added a new section 7.5 in the Discussion section addressing this point and providing some theoretical estimations of the crucial adsorbed layer thicknesses for certain practical membrane systems.
Point 2: We added section 9 "Nomenclature" listing the major parameters.
Point 3: We introduced minor changes on pp. 2, 8-10 to more clearly distinguish between the known and the new results.
Reviewer 2 Report
The present work provides a very comprehensive theoretical analysis of the adsorption of semiflexible polymer chains on permeable planar membranes whereby the authors employ both scaling analysis as well as transfer matrix methods in order to present a rather quantitative description of the phenomenon. The linear semirigid macromolecules are considered within the Kratky-Porod model of wormlike chains which is widely accepted as a proper phenomenological model of semiflexible polymers. The study considers two models of a substrate: (i) a perfectly permeable fluid layer with an attractive short-range potential, and (ii) a porous solid thin film with holes, characterized by porosity w. Among the most important results one can point out the predicted structure of the adsorbed layers in the vicinity and far from the membrane and the derived functional dependence c(x) of the monomers' density at distance x from the substrate, which in model (i) is governed by an exponent \beta =1 (in contrast to 4/3 for impenetrable solid substrates) while this critical exponent is found to depend on the porosity w in the second model. The paper is clearly written with appropriate figures and is sufficiently concise albeit rather long. It will certainly be of strong interest for the broad community of researchers in the area of soft condensed matter and biophysics. Therefore, I recommend the publication of this work in the journal of polymers as is.
As a caveat one should note, however, that the consideration of the second model of porous substrate is less clear and poses a number of open questions. For example, it would have been desirable to see some analytic results regarding the functional form for the probability distribution of train sizes (which is exponential for impenetrable substrates) and the way the trains depend on the porosity w, e.g,. when w ---> 1. Moreover, given that the holes exert themselves no attraction on the monomers [i.e., u* = u*(w)], one would expect that at least at distance \Delta from the rim of each hole the adsorption strength is going to be additionally reduced which would probably renormalize w further and may affect the entire picture.
Author Response
We thank the referee for careful reading of our manuscript and useful points made. Our response is given below:
--- We indicated more clearly the conditions on the hole diameter D and the degree of porosity w adopted in our theory (see section 6, pp. 16, 17 and the new Equations 83, 84).
--- We added a new section 7.2 specifying an important condition (Equation 83) defining the lower bound for the loop length in the self-similar sublayer. The train size distribution and the effect of porosity for it are discussed in the second part of section 7.2. In particular, Equation 84 provides a sufficient condition for the train size distribution to remain nearly exponential.